# Learning Long Range Dependencies on Graphs via Random Walks

**Dexiong Chen, Till Hendrik Schulz & Karsten Borgwardt**
Max Planck Institute of Biochemistry, 82152 Martinsried, Germany
{dchen, tschulz, borgwardt}@biochem.mpg.de

## Abstract

Message-passing graph neural networks (GNNs) excel at capturing local relationships but struggle with long-range dependencies in graphs. In contrast, graph transformers (GTs) enable global information exchange but often oversimplify the graph structure by representing graphs as sets of fixed-length vectors. This work introduces a novel architecture that overcomes the shortcomings of both approaches by combining the long-range information of random walks with local message passing. By treating random walks as sequences, our architecture leverages recent advances in sequence models to effectively capture long-range dependencies within these walks. Based on this concept, we propose a framework that offers (1) more expressive graph representations through random walk sequences, (2) the ability to utilize any sequence model for capturing long-range dependencies, and (3) the flexibility by integrating various GNN and GT architectures. Our experimental evaluations demonstrate that our approach achieves competitive performance on 19 graph and node benchmark datasets, notably outperforming existing methods by up to 13% on the PascalVoc-SP and COCO-SP datasets.

## 1 Introduction

Message-passing graph neural networks (GNNs) (Gilmer et al., 2017) and graph transformers (GTs) (Ying et al., 2021), have emerged as powerful tools for learning on graphs. While GNNs are efficient in identifying local relationships, they often fail to capture distant interactions due to the local nature of message passing, leading to issues such as over-smoothing (Oono & Suzuki, 2020) and over-squashing (Alon & Yahav, 2021). In contrast, GTs (Ying et al., 2021; Mialon et al., 2021; Chen et al., 2022a; Rampášek et al., 2022; Shirzad et al., 2023) address these limitations by directly modeling long-range interactions through global attention mechanisms, enabling information exchange between all nodes. However, GTs typically preprocess the complex graph structure into fixed-length vectors for each node, using positional or structural encodings (Rampášek et al., 2022). This approach essentially treats the graph as a set of nodes enriched with these vectors. Such vector representations of graph topologies inevitably result in a loss of structural information, limiting expressivity even when GTs are combined with local message-passing techniques (Zhu et al., 2023). In this work, we address these limitations by introducing a novel architecture that captures long-range dependencies while preserving rich structural information, by leveraging the power of random walks.

Random walks offer a flexible approach to exploring graphs, surpassing the limitations of fixed-length vector representations. By traversing diverse paths across the graph, random walks can capture subgraphs with large diameters, such as cycles, which message passing often struggles to represent, due to its depth-first nature (Grover & Leskovec, 2016). More importantly, *the complexity of sampling random walks is determined by their length and sample size rather than the overall size of the graph*. This characteristic makes random walks a scalable choice for representing large graphs, offering clear computational advantages compared to many computationally expensive encoding methods.

While several graph learning approaches have employed random walks, their full potential remains largely untapped. Most existing approaches either focus solely on short walks (Chen et al., 2020; Nikolentzos & Vazirgiannis, 2020) or use walks primarily for structural encoding, neglecting the rich information they contain (Dwivedi et al., 2021; Mialon et al., 2021). A more recent method, CRaWL (Tönshoff et al., 2023b), takes a novel approach by representing a graph as a set of random

walks. While this approach shows promising results, it has two major practical limitations: 1) its reliance on convolutional layers to process random walks, particularly with small kernel sizes, constrains its ability to approximate arbitrary functions on walks and fully capture long-range dependencies within each walk. 2) Due to the depth-first nature of random walks, it struggles to efficiently capture local relationships, such as simple subtrees, as illustrated in Figure 1.

Considering the limitations of existing random-walk-based models, we propose an approach that leverages the strengths of two complementary graph exploration paradigms. Our method combines the local neighborhood information captured by the *breadth-first nature* of message passing with the long-range dependencies obtained through the *depth-first* nature of random walks. Unlike GTs that encode random walks into fixed-length vectors (Rampášek et al., 2022; Chen et al., 2022a), our approach preserves their sequential nature, thereby retaining richer structural information. Our proposed

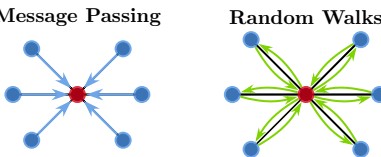

Figure 1: Message passing efficiently captures locally sparse subgraphs, like $k$-star subgraphs, while random walks struggle, requiring a length of $2k$.

architecture, named *NeuralWalker*, achieves this by processing sets of sampled random walks using powerful sequence models. We then employ local (and optionally global) message passing to capture complementary information. Multiple alternations of these two operations are stacked to form our model. A key innovation of our approach is the utilization of long sequence models, such as state space models, to learn from random walk sequences.

Our contributions are summarized as follows. i) We propose a novel framework that leverages both random walks and message passing, leading to provably more expressive graph representation. ii) Our model exploits advances in sequence modeling (*e.g,* transformers and state space models) to capture long-range dependencies within the walks. iii) Our message-passing block can seamlessly integrate various GNN and GT architectures, allowing for customization based on specific tasks. iv) We conduct extensive ablation studies to offer practical insights for choosing the optimal sequence layer types and message-passing strategies. Notably, the trade-off between model complexity and expressivity can be flexibly controlled by adjusting walk sampling rate and length, making our model scalable to graphs with up to 1.6M nodes. v) Our model demonstrates remarkable performance improvements over existing methods on a comprehensive set of 19 benchmark datasets.

## 2 RELATED WORK

**Local and global message passing.** Message-passing neural networks (MPNNs) are a cornerstone of graph learning. They propagate information between nodes, categorized as either local or global methods based on the propagation range. Local MPNNs, also known as GNNs (*e.g,* GCN (Kipf & Welling, 2016), GIN (Xu et al., 2019)), excel at capturing local relationships but struggle with distant interactions due to limitations like over-smoothing (Oono & Suzuki, 2020) and over-squashing (Alon & Yahav, 2021). Global message passing offers a solution by modeling long-range dependencies through information exchange across all nodes. GTs (Ying et al., 2021; Kreuzer et al., 2021; Mialon et al., 2021; Chen et al., 2022a; Rampášek et al., 2022; Shirzad et al., 2023), using global attention mechanisms, are a prominent example. However, GTs achieve this by compressing the graph structure into fixed-length vectors, leading to a loss of rich structural information. Alternative techniques include the virtual node approach (Gilmer et al., 2017; Barceló et al., 2020), which enables information exchange between distant nodes by introducing an intermediary virtual node.

**Random walks for graph learning.** Random walks have a long history in graph learning, particularly within traditional graph kernels. Due to the computational intractability of subgraph or path kernels (Gärtner et al., 2003), walk kernels (Gärtner et al., 2003; Kashima et al., 2003; Borgwardt & Kriegel, 2005) were introduced to compare common walks or paths in two graphs efficiently. Non-backtracking walks have also been explored (Mahé et al., 2005) for molecular graphs. In deep graph learning, several approaches utilize walks or paths to enhance GNN expressivity. GCKN (Chen et al., 2020) pioneered short walk and path feature aggregation within graph convolution, further explored in Michel et al. (2023). RWGNN (Nikolentzos & Vazirgiannis, 2020) leverages differentiable walk kernels for subgraph comparison and parametrized anchor graphs. The closest work to ours is CRaWL (Tönshoff et al., 2023b). However, it lacks message passing and relies on a convolutional layer, particularly with small kernel sizes, limiting its universality. Prior to CRaWL,

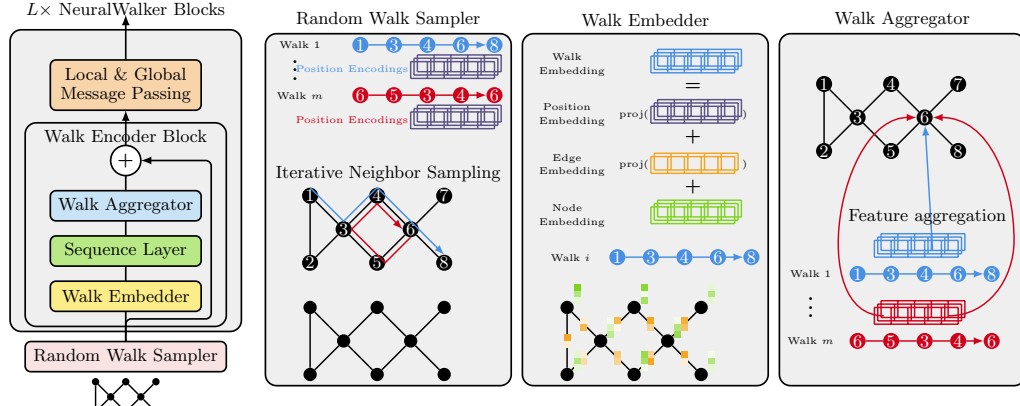

Figure 2: Overview of the NeuralWalker architecture. The random walk sampler samples $m$ random walks independently without replacement; the walk embedder computes walk embeddings given the node/edge embeddings at the current layer; the walk aggregator aggregates walk features into the node features via pooling of the node features encountered in all the walks passing through the node.

Ivanov & Burnaev (2018); Wang et al. (2021); Yin et al. (2022) used random walks with identity encodings, which served as inspiration for CRaWL. However, these approaches did not include adjacency encodings, which are essential for capturing the full information of induced subgraphs. Additionally, random walks have been used as structural encoding in GTs such as RWPE (Dwivedi et al., 2021) and relative positional encoding in self-attention (Mialon et al., 2021).

**Sequence modeling.** Sequence models, particularly transformers (Vaswani et al., 2017) and state space models (SSMs) (Gu et al., 2021; Gu & Dao, 2023), have become instrumental in natural language processing (NLP) and audio processing due to their ability to capture long-range dependencies within sequential data. However, directly leveraging these models on graphs remains challenging due to the inherent structural differences. Existing approaches like GTs treat graphs as sets of nodes, hindering the application of transformer architectures to sequences within the graph. Similarly, recent work utilizing SSMs for graph modeling (Wang et al., 2024) relies on node orderings based on degrees, a suboptimal strategy that may introduce biases or artifacts when creating these artificial sequences that do not reflect the underlying graph topology. Behrouz & Hashemi (2024) use SSMs on nested sequences of subgraphs induced by random walks of growing sizes and redundant encodings (*e.g*, Laplacian PEs and relative PE/SEs), introducing additional complexity and limiting walk lengths.

Our work addresses this limitation by explicitly treating random walks on graphs as sequences. This allows us to leverage the power of state-of-the-art (SOTA) sequence models to capture rich structural information within these walks, ultimately leading to a more universal graph representation. Furthermore, by integrating both message passing and random walks, our model is provably more expressive compared to existing MPNNs and random walk-based models, as discussed in Section 4.

## 3 NEURAL WALKER

In this section, we present the architecture of our proposed NeuralWalker, which processes sequences obtained from random walks to produce both node and graph representations. Its components consist of a random walk sampler, described in Section 3.2, and a stack of NeuralWalker blocks, discussed in Section 3.3. A visualization of the architecture can be found in Figure 2.

### 3.1 NOTATION AND RANDOM WALKS ON GRAPHS

We first introduce the necessary notation. A graph is a tuple $G = (V, E, x, z)$, where $V$ is the set of nodes, $E$ is the set of edges, and $x : V \to \mathbb{R}^d$ and $z : E \to \mathbb{R}^{d'}$ denote the functions assigning attributes to node and edges, respectively. We denote by $\mathcal{G}$ and $\mathcal{G}_n$ the space of all graphs and the space of graphs up to size $n$, respectively. The neighborhood of a node $v$ is denoted by $\mathcal{N}(v)$ and its degree by $d(v)$. A walk $W$ of length $\ell$ on a graph $G$ is a sequence of nodes connected by edges, i.e. $W = (w_0, \dots, w_\ell) \in V^{\ell+1}$ such that $w_{i-1} w_i \in E$ for all $i \in [\ell]$. We denote by $\mathcal{W}(G)$ and $\mathcal{W}_\ell(G)$ the set of all walks and all walks of length $\ell$ on $G$, respectively. $W$ is called non-backtracking if

$w_{i-1} \neq w_{i+1}$ for all $i$ and we denote the set of all such walks by $\mathcal{W}_\ell^{\mathrm{nb}}(G)$. A random walk is a Markov chain that starts with some distribution on nodes $P_0(v)$ and transitions correspond to moving to a neighbor chosen uniformly at random. For non-backtracking random walks, neighbors are chosen uniformly from $\mathcal{N}(w_i)\backslash\{w_{i-1}\}$. We denote by $P(\mathcal{W}(G), P_0)$ the distribution of random walks with initial distribution $P_0$, and by $P(\mathcal{W}(G))$ the case where $P_0 = \mathbb{U}(V)$ is the uniform distribution on $V$.

## 3.2 RANDOM WALK SAMPLER

The random walk sampler independently samples a subset of random walks on each graph through a probability distribution on all possible random walks. For any distribution on random walks $P(\mathcal{W}(G), P_0)$, we denote by $P_m(\mathcal{W}(G)) := \{W_1, \ldots, W_m\}$ a realization of $m$ i.i.d. samples $W_j \sim P(\mathcal{W}(G), P_0)$. Our model always operates on such realizations. Motivated by the success in Tönshoff et al. (2023b), we consider non-backtracking walks of fixed length. Specifically, we consider the uniform distribution of length-$\ell$ random walks $P(\mathcal{W}(G), P_0) := P(\mathcal{W}_\ell^{\mathrm{nb}}(G), \mathbb{U}(V))$.

In practice, we restrict the number of samples $m \leq n$ where $n = |V|$ for computation efficiency. We define the sampling rate of random walks as the ratio of random walks to nodes ($\gamma := {}^m/n$). Note that random walks only need to be sampled once for each forward pass and that an efficient CPU implementation can be achieved through iterative neighbor sampling, *with a complexity $O(n\gamma\ell)$, linear in the number and length of random walks*. We remark that during inference, a higher sampling rate than that used during training can be used to enhance performance. Therefore, we always fix it to 1.0 at inference. In Section 5.3, we empirically study the impact of $\gamma$ and $\ell$ used at training on the performance, showing that once these hyperparameters exceed a certain threshold, their impact on performance saturates. We present below the positional encodings for random walks encoding full information of induced subgraphs along the walks, essential for establishing our theoretical results.

**Positional encodings for random walks.** Similar to Tönshoff et al. (2023b), we utilize additional encoding features that store connectivity information captured within random walks. In particular, we consider an identity encoding which encodes whether two nodes in a walk are identical within a window and an adjacency encoding which includes information about subgraphs induced by nodes along the walk. Specifically, for a walk $W = (w_0, \ldots, w_\ell) \in \mathcal{W}_\ell(G)$ and window size $s \in \mathbb{N}_+$, the identity encoding $W$, denoted $\mathrm{id}_W^s$, is the binary matrix in $\{0, 1\}^{(\ell+1)\times s}$ with $\mathrm{id}_W^s[i, j] = 1$ if $w_i = w_{i-j-1}$ s.t. $i - j \geq 1$, and otherwise 0 for any $0 \leq i \leq \ell$ and $0 \leq j \leq s - 1$. Similarly, the adjacency encoding $\mathrm{adj}_W^s \in \{0, 1\}^{(\ell+1)\times(s-1)}$ satisfies $\mathrm{adj}_W^s[i, j] = 1$ if $w_i w_{i-j-1} \in E$ s.t. $i - j \geq 1$, and otherwise 0 for any $0 \leq i \leq \ell$ and $0 \leq j \leq s - 1$. A visual example of such encodings is given in Appendix B.1. Finally, the output of the random walk sampler is the concatenation all encodings into a single matrix $h_{\mathrm{pe}} \in \mathbb{R}^{(\ell+1)\times d_{\mathrm{pe}}}$ together with the sampled random walks.

## 3.3 MODEL ARCHITECTURE

In the following, we describe the architecture of NeuralWalker which conists of several walk encoder blocks where each block is comprised of three components: a walk embedder, a sequence layer, and a walk aggregator that are presented in Sections 3.3.1, 3.3.2, and 3.3.3, respectively.

### 3.3.1 WALK EMBEDDER

The walk embedder generates walk embeddings given the sampled walks, and the node and edge embeddings at the current layer. It is defined as a function $f_{\mathrm{emb}} : \mathcal{W}_\ell(G) \to \mathbb{R}^{(\ell+1)\times d}$. Specifically, for any walk $W \in P_m(\mathcal{W}_\ell(G))$, the walk embedding $h_W := f_{\mathrm{emb}}(W) \in \mathbb{R}^{(\ell+1)\times d}$ is defined as

$$h_W[i] := h_V(w_i) + \mathrm{proj}_{\mathrm{edge}}(h_E(w_i w_{i+1})) + \mathrm{proj}_{\mathrm{pe}}(h_{\mathrm{pe}}[i]), \tag{1}$$

where $h_V : V \to \mathbb{R}^d$ and $h_E : E \to \mathbb{R}^{d_{\mathrm{edge}}}$ are node and edge embeddings at the current block and $\mathrm{proj}_{\mathrm{edge}} : \mathbb{R}^{d_{\mathrm{edge}}} \to \mathbb{R}^d$ and $\mathrm{proj}_{\mathrm{pe}} : \mathbb{R}^{d_{\mathrm{pe}}} \to \mathbb{R}^d$ are some trainable projection functions. The resulting walk embeddings is then processed with a sequence model as discussed below.

### 3.3.2 SEQUENCE LAYER ON WALK EMBEDDINGS

In principle, any sequence model can be used to process the walk embeddings obtained above. A sequence layer transforms a sequence of feature vectors into a new sequence, *i.e.,* it is a function $f_{\mathrm{seq}} : \mathbb{R}^{(\ell+1)\times d} \to \mathbb{R}^{(\ell+1)\times d}$. In the following, we discuss several choices for such a function.

1D CNNs are simple and fast models for processing sequences, also used in Tönshoff et al. (2023b). They are GPU-friendly and require relatively limited memory. However, the receptive field of a 1D CNN is limited by its kernel size, which might fail to capture distant dependencies on long walks.

Transformers are widely used in modeling sequences and graphs due to their universality and strong performance. However, we found in our experiments (see Table 6) that they are suboptimal encoders for walk embeddings, even when equipped with the latest techniques like RoPE (Su et al., 2024).

SSMs are a more recent approach for modeling long sequences. In our experiments, we employ two of the latest instances of SSMs, namely S4 (Gu et al., 2021) and Mamba (Gu & Dao, 2023). In addition to the original version, we consider the bidirectional version of Mamba (Zhu et al., 2024). We found that bidirectional Mamba consistently outperforms other options (Section 5.3). For a more comprehensive background on SSMs, please refer to Appendix A.3.

### 3.3.3 WALK AGGREGATOR

The walk aggregator aggregates walk features into node features such that the resulting node features encode context information from all walks passing through that node. It is defined as a function $f_{\text{agg}} : (P_m(\mathcal{W}_\ell(G)) \to \mathbb{R}^{(\ell+1)\times d}) \to (V \to \mathbb{R}^d)$ and the resulting node feature mapping is given by $h_V^{\text{agg}} := f_{\text{agg}}(f_{\text{seq}}(f_{\text{emb}}|_{P_m(\mathcal{W}_\ell(G))}))$ where $f|.$ denotes the function restriction. In this work, we consider the average of all the node features encountered in the walks passing through a given node. Specifically, the node feature mapping $h_V^{\text{agg}}$ with an average pooling is defined as

$$h_V^{\text{agg}}(v) := \frac{1}{N_v(P_m(\mathcal{W}_\ell(G)))} \sum_{W \in P_m(\mathcal{W}_\ell(G))} \sum_{w_i \in W \, st. \, w_i = v} f_{\text{seq}}(h_W)[i], \qquad (2)$$

where $N_v(P_m(\mathcal{W}_\ell(G)))$ represents the number of occurrences of $v$ in the union of walks in $P_m(\mathcal{W}_\ell(G))$. One could also average the edge features in the walks passing through a certain edge to update the edge features: $h_E^{\text{agg}}(e) := \sum_{W \in P_m(\mathcal{W}_\ell(G))} \sum_{w_i w_{i+1} \in W \, st. \, w_i w_{i+1} = e} f_{\text{seq}}(h_W)[i]$ up to a normalization factor. In practice, we also use skip connections to keep track of the node features from previous layers.

### 3.3.4 LOCAL AND GLOBAL MESSAGE PASSING

While random walks are efficient at identifying long-range dependencies due of their depth-first nature, they are less suited for capturing local substructure information, which often plays an essential role in many graph learning tasks. To address this limitation, we draw inspiration from classic node embedding methods (Perozzi et al., 2014; Grover & Leskovec, 2016). We incorporate a message-passing layer into our encoder block, leveraging its breadth-first characteristics to complement the information obtained through random walks. Such a (local) message passing step is given by

$$h_V^{\text{mp}}(v) := h_V^{\text{agg}}(v) + \text{MPNN}(G, h_V^{\text{agg}}(v)), \qquad (3)$$

where MPNN denotes a GNN model, typically with one layer in each encoder block.

Following the local message passing layer, we optionally apply a global message passing mechanism, allowing for a global information exchange, as done in GTs (Chen et al., 2022a). We particularly consider two global message passing techniques, namely virtual node (Gilmer et al., 2017; Tönshoff et al., 2023a) and transformer layer (Ying et al., 2021; Chen et al., 2022a; Rampášek et al., 2022). We provide more details on these techniques in Appendix B.2.

## 4 THEORETICAL RESULTS

In this section, we investigate the theoretical properties of NeuralWalker. The proofs of the following results as well as more background can be found in Appendix C.

We first define walk feature vectors following Tönshoff et al. (2023b):

**Definition 4.1** (Walk feature vector). *For any graph $G = (V, E, x, z)$ and $W \in \mathcal{W}_\ell(G)$, the walk feature vector $X_W$ of $W$ is defined by concatenating the node and edge feature vectors, along with the positional encodings of $W$ with window size $s = \ell$. Formally,*

$$X_W = (x(w_i), z(w_i w_{i+1}), h_{pe}[i])_{i=0,\dots,\ell} \in \mathbb{R}^{(\ell+1)\times d_{walk}},$$

where $h_{pe}$ represents the positional encoding of Section 3.2, $z(w_\ell w_{\ell+1}) = 0$, and $d_{walk} := d+d'+d_{pe}$. *For simplicity, we still denote the distribution of walk feature vectors on $G$ by $P(\mathcal{W}(G))$.*

For simplicity, we consider general rather than non-backtracking random walks in this section. Now assume that we apply an average pooling followed by a linear layer to the output of the walk aggregator in Eq. (2). By adjusting the normalization factor to a constant $m\ell/|V|$, we can express the function $g_{f,m,\ell}$ on $\mathcal{G}$ as an average over functions of walk feature vectors:

$$g_{f,m,\ell}(G) = \frac{1}{m} \sum_{W \in P_m(\mathcal{W}_\ell(G))} f(X_W) \tag{4}$$

where $f : \mathbb{R}^{d_{walk}} \to \mathbb{R}$ is some function on walk feature vectors.

If we sample a sufficiently large number of random walks, the average function $g_{f,m,\ell}(G)$ converges almost surely to $g_{f,\ell} := \mathbb{E}_{X_W \sim P(\mathcal{W}_\ell(G))}[f(X_W)]$, due to the law of large numbers. This result can be further quantified using the central limit theorem, which provides a rate of convergence (see Theorem C.5 in Appendix). Furthermore, we have the following useful properties of this limit:

**Theorem 4.2** (Lipschitz continuity)**.** *For some functional space $\mathcal{F}$ of functions on walk feature vectors, we define the following distance $d_\mathcal{F} : \mathcal{G} \times \mathcal{G} \to \mathbb{R}_+$:*

$$d_{\mathcal{F},\ell}(G, G') := \sup_{f \in \mathcal{F}} \left| \mathbb{E}_{X_W \sim P(\mathcal{W}_\ell(G))}[f(X_W)] - \mathbb{E}_{X_{W'} \sim P(\mathcal{W}_\ell(G'))}[f(X_{W'})] \right|. \tag{5}$$

*Then $(\mathcal{G}_n, d_{\mathcal{F},\ell})$ is a metric space if $\mathcal{F}$ is a universal space and $\ell \geq 4n^3$.*

*If $\mathcal{F}$ contains $f$, then for any $G, G' \in \mathcal{G}_n$, we have*

$$|g_{f,\ell}(G) - g_{f,\ell}(G')| \leq d_{\mathcal{F},\ell}(G, G'). \tag{6}$$

*In particular, if $f \in \mathcal{F}$ is an L-Lipschitz function, the difference in outputs is bounded by the earth mover's distance $W_1(\cdot, \cdot)$ between the distributions of walk feature vectors:*

$$|g_{f,\ell}(G) - g_{f,\ell}(G')| \leq L \cdot W_1(P(\mathcal{W}_\ell(G)), P(\mathcal{W}_\ell(G'))). \tag{7}$$

The Lipschitz constant, widely used to assess neural network stability under small perturbations (Virmaux & Scaman, 2018), guarantees that NeuralWalker maintains stability when subjected to minor alterations in graph structure. Notably, parameterizing $f$ with several neural network layers yields a Lipschitz constant comparable to that of MPNNs on a pseudometric space defined by the tree mover's distance (Chuang & Jegelka, 2022). However, a key distinction lies in the input space metrics: while MPNNs operate on tree structures, NeuralWalker focuses on the distribution of walk feature vectors. A more comprehensive comparison of MPNNs' and NeuralWalker's stability and generalization under distribution shift is left for future research.

**Theorem 4.3** (Injectivity)**.** *Assume that $\mathcal{F}$ is a universal space. If $G$ and $G'$ are non-isomorphic graphs, then there exists an $f \in \mathcal{F}$ such that $g_{f,\ell}(G) \neq g_{f,\ell}(G')$ if $\ell \geq 4\max\{|V|, |V'|\}^3$.*

The injectivity property ensures that our model with a sufficiently large number of sufficiently long ($\geq 4n^3$) random walks can distinguish between non-isomorphic graphs. It is worth noting that although our assumptions include specific conditions on the random walk length to establish the space as a metric, removing the length constraint still results in a pseudometric space. In this case, $d_{\mathcal{F},\ell}(G, G') > 0$ if $G$ and $G'$ are distinguishable by the $(\lfloor \ell/2 \rfloor + 1)$-subgraph isomorphism test, where $\lfloor \cdot \rfloor$ is the floor function (*i.e.,* they do not have the same set of subgraphs up to size $(\lfloor \ell/2 \rfloor + 1)$).

Using the previous result jointly with the message-passing module, we arrive at the following result, which particularly highlights the advantage of combining random walks and message passing.

**Theorem 4.4.** *For any $\ell \geq 2$, NeuralWalker equipped with the complete walk set $\mathcal{W}_\ell$ is strictly more expressive than 1-WL and the $(\lfloor \ell/2 \rfloor + 1)$-subgraph isomorphism test, and thus ordinary MPNNs.*

The injectivity in Thm. 4.3 is guaranteed only if $\mathcal{F}$ is a universal functional space. This condition highlights a limitation in approaches like CRaWL (Tönshoff et al., 2023b) which employs CNNs to process walk feature vectors. CNNs can only achieve universality under strict conditions, including periodic boundary conditions and a large number of layers (Yarotsky, 2022). However, random walks generally do not satisfy periodic boundary conditions, and utilizing an excessive number of

Table 1: Test performance on benchmarks from Dwivedi et al. (2023). Metrics with mean ± std of 4 runs are reported. The result with ⋆ is obtained using the pretraining strategy presented in Section 5.2.

| | ZINC | MNIST | CIFAR10 | PATTERN | CLUSTER |
|---|---|---|---|---|---|
| # GRAPHS | 12K | 70K | 60K | 14K | 12K |
| AVG. # NODES | 23.2 | 70.6 | 117.6 | 118.9 | 117.2 |
| AVG. # EDGES | 24.9 | 564.5 | 941.1 | 3039.3 | 2150.9 |
| METRIC | MAE ↓ | ACC ↑ | ACC ↑ | ACC ↑ | ACC ↑ |
| GCN | 0.367 ± 0.011 | 90.705 ± 0.218 | 55.710 ± 0.381 | 71.892 ± 0.334 | 68.498 ± 0.976 |
| GIN | 0.526 ± 0.051 | 96.485 ± 0.252 | 55.255 ± 1.527 | 85.387 ± 0.136 | 64.716 ± 1.553 |
| GATEDGCN | 0.282 ± 0.015 | 97.340 ± 0.143 | 67.312 ± 0.311 | 85.568 ± 0.088 | 73.840 ± 0.326 |
| SAT | 0.089 ± 0.002 | – | – | 86.848 ± 0.037 | 77.856 ± 0.104 |
| GPS | 0.070 ± 0.004 | 98.051 ± 0.126 | 72.298 ± 0.356 | 86.685 ± 0.059 | 78.016 ± 0.180 |
| EXPHORMER | – | 98.55 ± 0.03 | 74.69 ± 0.13 | 86.70 ± 0.03 | 78.07 ± 0.037 |
| GRIT | 0.059 ± 0.002 | 98.108 ± 0.111 | 76.468 ± 0.881 | **87.196 ± 0.076** | **80.026 ± 0.277** |
| GRED | 0.077 ± 0.002 | 98.383 ± 0.012 | 76.853 ± 0.185 | 86.759 ± 0.020 | 78.495 ± 0.103 |
| GMN | – | 98.39 ± 0.18 | 75.76 ± 0.42 | 87.14 ± 0.12 | – |
| CRAWL | 0.085 ± 0.004 | 97.944 ± 0.050 | 69.013 ± 0.259 | – | – |
| NEURALWALKER | **0.053 ± 0.002**⋆ | **98.692 ± 0.079** | **76.903 ± 0.457** | 86.977 ± 0.012 | 78.189 ± 0.188 |

Table 2: Test performance on LRGB (Dwivedi et al., 2022). Metrics with mean ± std of 4 runs are reported. NeuralWalker improves the best baseline by 10% and 13% on **PascalVOC-SP** and **COCO-SP** respectively. GPS-tuned refers to the results reported by Tönshoff et al. (2023a) with a more extensive hyperparameter tuning compared to GPS (Rampášek et al., 2022).

| | PASCALVOC-SP | COCO-SP | PEPTIDES-FUNC | PEPTIDES-STRUCT | PCQM-CONTACT |
|---|---|---|---|---|---|
| # GRAPHS | 11.4K | 123.3K | 15.5K | 15.5K | 529.4K |
| AVG. # NODES | 479.4 | 476.9 | 150.9 | 150.9 | 30.1 |
| AVG. # EDGES | 2,710.5 | 2,693.7 | 307.3 | 307.3 | 61.0 |
| METRIC | F1 ↑ | F1 ↑ | AP ↑ | MAE ↓ | MRR ↑ |
| GCN | 0.2078 ± 0.0031 | 0.1338 ± 0.0007 | 0.6860 ± 0.0050 | 0.2460 ± 0.0007 | 0.4526 ± 0.0006 |
| GIN | 0.2718 ± 0.0054 | 0.2125 ± 0.0009 | 0.6621 ± 0.0067 | 0.2473 ± 0.0017 | 0.4617 ± 0.0005 |
| GATEDGCN | 0.3880 ± 0.0040 | 0.2922 ± 0.0018 | 0.6765 ± 0.0047 | 0.2477 ± 0.0009 | 0.4670 ± 0.0004 |
| GPS | 0.3748 ± 0.0109 | 0.3412 ± 0.0044 | 0.6535 ± 0.0041 | 0.2500 ± 0.0005 | – |
| GPS-TUNED | 0.4440 ± 0.0065 | 0.3884 ± 0.0055 | 0.6534 ± 0.0091 | 0.2509 ± 0.0014 | 0.4703 ± 0.0014 |
| EXPHORMER | 0.3975 ± 0.0037 | 0.3455 ± 0.0009 | 0.6527 ± 0.0043 | 0.2481 ± 0.0007 | – |
| GRIT | – | – | 0.6988 ± 0.0082 | 0.2460 ± 0.0012 | – |
| GRED | – | – | **0.7133 ± 0.0011** | **0.2455 ± 0.0013** | – |
| GMN | 0.4393 ± 0.0112 | 0.3974 ± 0.0101 | 0.7071 ± 0.0083 | 0.2473 ± 0.0025 | – |
| GRAPH-MAMBA | 0.4191 ± 0.0126 | 0.3960 ± 0.0175 | 0.6739 ± 0.0087 | 0.2478 ± 0.0016 | – |
| CRAWL | – | – | 0.7074 ± 0.0032 | 0.2506 ± 0.0022 | – |
| NEURALWALKER | **0.4912 ± 0.0042** | **0.4398 ± 0.0033** | 0.7096 ± 0.0078 | 0.2463 ± 0.0005 | **0.4707 ± 0.0007** |

layers can exacerbate issues such as over-squashing and over-smoothing. In contrast, the sequence models considered in this work, such as transformers and SSMs, are universal approximators for any sequence-to-sequence functions (Yun et al., 2020; Wang & Xue, 2024). Furthermore, the proof of Thm. 4.4 suggests that random walk-based models without message passing cannot be more expressive than 1-WL. Consequently, our model is provably more expressive than CRaWL.

Finally, we have the following complexity results:

**Theorem 4.5** (Complexity). *The complexity of NeuralWalker, when used with Mamba (Gu & Dao, 2023), is $O(kdn(\gamma\ell + \beta))$, where $k, d, n, \gamma, \ell, \beta$ denote the number of layers, hidden dimensions, the (maximum) number of nodes, sampling rate, length of random walks and average degree, respectively.*

## 5 EXPERIMENTS

In this section, we compare NeuralWalker to several SOTA models on a diverse set of 19 benchmark datasets. Furthermore, we provide a detailed ablation study on components of our model. Appendix D provides more details about the experimental setup, datasets, runtime, and additional results.

### 5.1 BENCHMARKING NEURALWALKER TO STATE-OF-THE-ART METHODS

We compare NeuralWalker against several popular message passing GNNs, GTs, and walk-based models. GNNs include GCN (Kipf & Welling, 2016), GraphSAGE (Hamilton et al., 2017), GIN (Xu et al., 2019), GAT (Veličković et al., 2018), GatedGCN (Bresson & Laurent, 2017). Models using global message passing include GraphTrans (Wu et al., 2021), SAT (Chen et al., 2022a), GPS (Rampášek et al., 2022), Exphormer (Shirzad et al., 2023), NAGphormer (Chen et al., 2022b), GRIT (Ma et al.,

2023), Polynormer (Deng et al., 2024), GRED (Ding et al., 2024), Graph-Mamba (Wang et al., 2024), GMN (Behrouz & Hashemi, 2024). Walk-based models include CRaWL (Tönshoff et al., 2023b). To ensure diverse benchmarking tasks, we use datasets from Benchmarking-GNNs (Dwivedi et al., 2023), Long-Range Graph Benchmark (LRGB) (Dwivedi et al., 2022), Open Graph Benchmark (OGB) (Hu et al., 2020a), and datasets from Platonov et al. (2022); Leskovec & Krevl (2014).

**Benchmarking GNNs.** We evaluated NeuralWalker's performance on five tasks from the Benchmarking GNNs suite: ZINC, MNIST, CIFAR10, PATTERN, and CLUSTER (results in Table 1). Notably, NeuralWalker achieved SOTA results on three out of five datasets and matched the best-performing model on the remaining two. While GRIT exhibited superior performance on the two small synthetic datasets, its scalability to larger datasets, such as those in LRGB, is limited, as demonstrated in the subsequent paragraph. It is worth noting that NeuralWalker significantly outperforms the previous SOTA random walk-based model, CRaWL. This improvement can be attributed to the integration of message passing and the Mamba architecture, as discussed in Sections 4. A more extensive empirical comparison of them is also given in Section 5.3. These results underscore NeuralWalker's robust performance across diverse synthetic benchmark tasks.

**Long-Range Graph Benchmark.** We further evaluated NeuralWalker's ability to capture long-range dependencies on the recently introduced LRGB benchmark, encompassing five datasets designed to test this very capability (details in Rampášek et al. (2022); Dwivedi et al. (2022)). Note that for PCQM-Contact, we used the filtered Mean Reciprocal Rank (MRR), introduced by Tönshoff et al. (2023a), as the evaluation metric. NeuralWalker consistently outperformed all baseline methods across all but two tasks (see Table 2). Notably, on PascalVOC-SP and COCO-SP, where previous work has shown the importance of long-range dependencies (*e.g.,* Tönshoff et al. (2023a)), NeuralWalker significantly surpassed the SOTA models by a substantial margin, *up to a 10% improvement.*

**Open Graph Benchmark.** To assess NeuralWalker's scalability on massive quantities of graphs, we evaluated it on the OGB benchmark, which includes datasets exceeding 100K graphs each. For computational efficiency, we employed 1D CNNs as the sequence layers in this experiment. NeuralWalker achieved SOTA performance on two out of the three datasets (Table 3), demonstrating its ability to handle large-scale graph data. However, the OGBG-PPA dataset presented challenges with overfitting. On this dataset, NeuralWalker with just one block outperformed its multiblock counterpart on this dataset, suggesting potential limitations in regularization needed for specific tasks.

**Node classification on large graphs.** We further explored NeuralWalker's ability to handle large graphs in node classification tasks. We integrated NeuralWalker with Polynormer (Deng et al., 2024), the current SOTA method in this domain. In this experiment, NeuralWalker utilized very long walks (up to 1,000 steps) with a low sampling rate ($\leq 0.01$) to capture long-range dependencies, replacing the transformer layer within Polynormer that still *struggles to scale to large graphs even with linear complexity.* Despite eschewing transformer layers entirely, NeuralWalker achieved performance comparable to Polynormer (Table 5), showing its scalability and effectiveness in modeling large graphs. Indeed, the complexity of NeuralWalker can be flexibly controlled by its sampling rate and length, as shown in Section 4. A notable highlight is NeuralWalker's ability to efficiently process the massive pokec dataset (1.6M nodes) using a single H100 GPU with 80GB of RAM.

## 5.2 Masked Positional Encoding Pretraining

Explicitly utilizing random walks as sequences offers a significant advantage: it allows for the application of advanced language modeling techniques. As a proof-of-concept, we adapt the BERT pretraining strategy (Devlin et al., 2019) to the positional encodings $h_{\text{pe}}$ of random walks. Our approach involves randomly replacing 15% of the positions in $h_{\text{pe}}$ with a constant vector of 0.5, with the objective of recovering the original binary encoding vectors for these positions. This method can be further enhanced by combining it with other established pretraining strategies, such as attributes masking (Hu et al., 2020b). Our experiments, as shown in Table 4, demonstrate that combining these strategies (*i.e.,* we first pretrain the model with masked positional encoding prediction and then continue with masked attributes pretraining) significantly enhances performance on the ZINC dataset.

## 5.3 Ablation studies

Here, we dissect the main components of our model architecture to gauge their contribution to predictive performance and to guide dataset-specific hyperparameter optimization.

Table 3: Test performance on OGB (Hu et al., 2020a). Metrics with mean ± std of 10 runs are reported.

| DATASET | OGBG-MOLPCBA | OGBG-PPA | OGBG-CODE2 |
|---|---|---|---|
| # GRAPHS | 437.9K | 158.1K | 452.7K |
| AVG. # NODES | 26.0 | 243.4 | 125.2 |
| AVG. # EDGES | 28.1 | 2,266.1 | 124.2 |
| METRIC | AP ↑ | ACC ↑ | F1 ↑ |
| GCN | 0.2424 ± 0.0034 | 0.6857 ± 0.0061 | 0.1595 ± 0.0018 |
| GIN | 0.2703 ± 0.0023 | 0.7037 ± 0.0107 | 0.1581 ± 0.0026 |
| GRAPHTRANS | 0.2761 ± 0.0029 | – | 0.1830 ± 0.0024 |
| SAT | – | 0.7522 ± 0.0056 | 0.1937 ± 0.0028 |
| GPS | 0.2907 ± 0.0028 | **0.8015 ± 0.0033** | 0.1894 ± 0.0024 |
| CRAWL | 0.2986 ± 0.0025 | – | – |
| NEURALWALKER | **0.3086 ± 0.0031** | 0.7888 ± 0.0059 | **0.1957 ± 0.0025** |

Table 4: Comparison of different pretraining strategies on the ZINC dataset. The pretraining was performed on ZINC *without using any external data*.

| STRATEGY | ZINC↓ |
|---|---|
| W/O PRETRAIN | 0.063 ± 0.001 |
| MASKED ATTR. | 0.061 ± 0.001 |
| MASKED PE | 0.055 ± 0.004 |
| MASKED PE + ATTR. | **0.053 ± 0.002** |

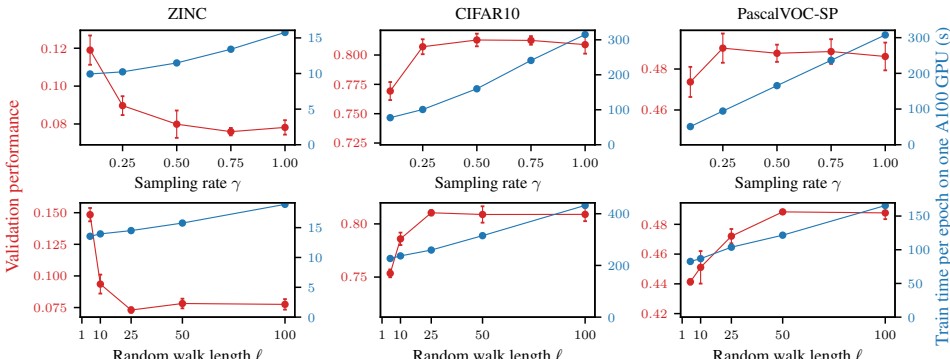

Figure 3: Validation performance when varying sampling rate and length of random walks.

We perform ablation studies on three datasets, from small to large graphs. Our analysis focuses on three key aspects: 1) We demonstrate the crucial role of integrating local and global message passing with random walks. 2) we evaluate various options for the sequence layer to identify the optimal choice. 3) We examine the impact of varying the sampling rate and length of random walks, revealing a trade-off between expressivity and computational complexity. Notably, *these parameters allow explicit control over model complexity*, a unique feature of our approach compared to subgraph MPNNs, which typically exhibit high complexity. All ablation experiments were performed on the *validation set*, with results averaged over four random seeds. The comprehensive findings are summarized in Table 6. Since NeuralWalker's output depends on the sampled random walks at inference, we demonstrate its robustness to sampling variability in Appendix D.5.

**Effect of local and global message passing.** Motivated by the limitations of the depth-first nature inherent in pure random walk-based encoders, as discussed in Section 3.3.4, this study investigates the potential complementary benefits of message passing. We conducted an ablation study (Table 6a) comparing NeuralWalker's variants with and without local or global message passing modules. For local message passing, we employed a GIN with edge features (Xu et al., 2019; Hu et al., 2019). Global message passing was explored using virtual node layers (Gilmer et al., 2017) and transformer layers (Vaswani et al., 2017; Chen et al., 2022a). Keeping the sequence layer fixed to Mamba, we observed that NeuralWalker with GIN consistently outperforms the version without, confirming the complementary strengths of random walks and local message passing. The impact of global message passing, however, varies across datasets, a phenomenon also noted by Rosenbluth et al. (2024). Interestingly, larger graphs like PascalVOC-SP demonstrate more significant gains from global message passing. This observation suggests promising directions for future research, such as developing methods to automatically identify optimal configurations for specific datasets.

**Comparison of sequence layer architectures.** We investigated the impact of various sequence layer architectures on walk embeddings, as shown in Table 6b. The architectures examined include CNN, transformer (with RoPE), and SSMs like S4 and Mamba. Surprisingly, transformers consistently underperformed compared to other architectures, contrasting with their good performance in other domains. This discrepancy may be attributed to the unique sequential nature of walk embeddings, which might not align well with the attention mechanism utilized by transformers.

Table 5: Test performance on node classification benchmarks from Platonov et al. (2022) and Leskovec & Krevl (2014). Metrics with mean ± std of 10 runs are reported.

| DATASET | ROMAN-EMPIRE | AMAZON-RATINGS | MINESWEEPER | TOLOKERS | QUESTIONS | POKEC |
|---|---|---|---|---|---|---|
| # NODES | 22,662 | 24,492 | 10,000 | 11,758 | 48,921 | 1,632,803 |
| # EDGES | 32,927 | 93,050 | 39,402 | 519,000 | 153,540 | 30,622,564 |
| METRIC | ACC ↑ | ACC ↑ | ROC AUC ↑ | ROC AUC ↑ | ROC AUC ↑ | ACC ↑ |
| GCN | 73.69 ± 0.74 | 48.70 ± 0.63 | 89.75 ± 0.52 | 83.64 ± 0.67 | 76.09 ± 1.27 | 75.45 ± 0.17 |
| GRAPHSAGE | 85.74 ± 0.67 | 53.63 ± 0.39 | 93.51 ± 0.57 | 82.43 ± 0.44 | 76.44 ± 0.62 | – |
| GAT(-SEP) | 88.75 ± 0.41 | 52.70 ± 0.62 | 93.91 ± 0.35 | 83.78 ± 0.43 | 76.79 ± 0.71 | 72.23 ± 0.18 |
| GPS | 82.00 ± 0.61 | 53.10 ± 0.42 | 90.63 ± 0.67 | 83.71 ± 0.48 | 71.73 ± 1.47 | OOM |
| NAGPHORMER | 74.34 ± 0.77 | 51.26 ± 0.72 | 84.19 ± 0.66 | 78.32 ± 0.95 | 68.17 ± 1.53 | 76.59 ± 0.25 |
| EXPHORMER | 89.03 ± 0.37 | 53.51 ± 0.46 | 90.74 ± 0.53 | 83.77 ± 0.78 | 73.94 ± 1.06 | OOM |
| POLYNORMER | 92.55 ± 0.37 | **54.81 ± 0.49** | 97.46 ± 0.36 | **85.91 ± 0.74** | **78.92 ± 0.89** | 86.10 ± 0.05 |
| NEURALWALKER | **92.92 ± 0.36** | 54.58 ± 0.36 | **97.82 ± 0.40** | 85.56 ± 0.74 | 78.52 ± 1.13 | **86.46 ± 0.09** |

Table 6: Ablation studies of NeuralWalker. Average validation performance over 4 runs is reported.

(a) Comparison of local and global message passing (MP). The sequence layer is fixed to Mamba. VN denotes the virtual node and Trans. denotes the transformer layer.

| MP (LOCAL + GLOBAL) | ZINC↓ | CIFAR10↑ | PASCALVOC-SP↑ |
|---|---|---|---|
| GIN + W/O | **0.085** | **80.885** | **0.4611** |
| W/O + W/O | 0.090 | 79.035 | 0.4525 |
| GIN + VN | **0.078** | 78.610 | 0.4672 |
| GIN + TRANS. | 0.083 | 80.755 | **0.4877** |
| GIN + W/O | 0.085 | **80.885** | 0.4611 |

(b) Comparison of sequence layers. Local and global MP are selected to give the best validation performance except for the highlighted row corresponding to CRaWL, which does not use message passing.

| SEQUENCE LAYER | ZINC↓ | CIFAR10↑ | PASCALVOC-SP↑ |
|---|---|---|---|
| MAMBA | **0.078** | **80.885** | **0.4877** |
| MAMBA (W/O BID) | 0.089 | 74.910 | 0.4522 |
| S4 | 0.082 | 77.970 | 0.4559 |
| CNN | 0.088 | 80.665 | 0.4652 |
| TRANS. | 0.084 | 72.850 | 0.4316 |
| CNN (W/O MP) | 0.116 | 78.760 | 0.3954 |

Mamba emerged as the top performer across all datasets, consistently outperforming its predecessors, S4 and the unidirectional version. However, CNNs present a compelling alternative for large datasets due to their faster computation (typically 2-3x faster than Mamba on A100). This presents a practical trade-off: Mamba offers superior accuracy but requires more computational resources. CNNs might be preferable for very large datasets or real-time applications where speed is critical. In our benchmarking experiments, we employed Mamba as the sequence layer, except for the OGB datasets. Finally, as predicted by Thm. 4.4, both our Mamba and CNN variants with message passing significantly outperform CRaWL which relies on CNNs and does not use any message passing.

**Impact of random walk sampling strategies.** We examined the impact of varying random walk sampling rates and lengths on NeuralWalker's performance, using Mamba as the sequence layer. While we adjusted the sampling rate during training, we fixed it at 1.0 for inference to maximize coverage. As anticipated, a larger number of longer walks led to improved coverage of the graph's structure, resulting in clear performance gains (Figure 3). However, this improvement plateaus as walks become sufficiently long, indicating diminishing returns beyond a certain threshold. Crucially, these performance gains come at the cost of increased computation time, which scales linearly with both sampling rate and walk length, as predicted by Thm. 4.5. This underscores the trade-off between expressivity and complexity, which can be explicitly controlled through these two hyperparameters. In practice, this trade-off between performance and computational cost necessitates careful consideration of resource constraints when selecting sampling rates and walk lengths. Future research could explore more efficient sampling strategies to minimize the necessary sampling rate.

## 6 CONCLUSION

We have introduced NeuralWalker, a powerful and flexible architecture that combines random walks and message passing to address the expressivity limitations of structural encoding in graph learning. By treating random walks as sequences and leveraging advanced sequence modeling techniques, NeuralWalker achieves superior performance compared to existing GNNs and GTs, as demonstrated through extensive experiments on various benchmarks. Looking forward, we acknowledge opportunities for further exploration. First, investigating more efficient random walk sampling strategies with improved graph coverage could potentially enhance NeuralWalker's performance. Second, exploring more self-supervised learning techniques for learning on random walks holds promise for extending NeuralWalker's applicability to unlabeled graphs.

## AVAILABILITY

Our code is available at https://github.com/BorgwardtLab/NeuralWalker.

## ACKNOWLEDGEMENTS

We thank Luis Wyss and Trenton Chang for their insightful feedback on the manuscript.

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

# Appendix

This appendix provides both theoretical and experimental materials and is organized as follows: Section A provides a more detailed background of related work. Section B presents some additional remarks on Neural Walker, including limitations and societal impacts. Section C provides theoretical background and proofs. Section D provides experimental details and additional results.

## A  BACKGROUND

### A.1  MESSAGE-PASSING GRAPH NEURAL NETWORKS

Graph Neural Networks (GNNs) refine node representations iteratively by integrating information from neighboring nodes. Xu et al. (2019) (Xu et al., 2019) provide a unifying framework for this process, consisting of three key steps: AGGREGATE, COMBINE, and READOUT. Various GNN architectures can be seen as variations within these functions.

In each layer, the AGGREGATE step combines representations from neighboring nodes (e.g., using sum or mean), which are then merged with the node's previous representation in the COMBINE step. This is typically followed by a non-linear activation function, such as ReLU. The updated representations are then passed to the next layer, and this process repeats for each layer in the network. These steps primarily capture local sub-structures, necessitating a deep network to model interactions across the entire graph.

The READOUT function ultimately aggregates node representations to the desired output granularity, whether at the node or graph level. Both AGGREGATE and READOUT steps must be permutation invariant. This framework offers a comprehensive perspective for understanding the diverse array of GNN architectures.

### A.2  TRANSFORMER ON GRAPHS

While Graph Neural Networks (GNNs) explicitly utilize graph structures, Transformers infer node relationships by focusing on node attributes. Transformers, introduced by Vaswani et al. (2017), treat the graph as a (multi-)set of nodes and use self-attention to determine node similarity.

A Transformer consists of two main components: a self-attention module and a feed-forward neural network (FFN). In self-attention, input features $\mathbf{X}$ are linearly projected into query ($\mathbf{Q}$), key ($\mathbf{K}$), and value ($\mathbf{V}$) matrices. Self-attention is then computed as:

$$\text{Attn}(\mathbf{X}) := \text{softmax}\left(\frac{\mathbf{Q}\mathbf{K}^T}{\sqrt{d_{\text{out}}}}\right)\mathbf{V} \in \mathbb{R}^{n \times d_{\text{out}}},$$

where $d_{\text{out}}$ is the dimension of $\mathbf{Q}$. Multi-head attention, which concatenates multiple instances of this equation, has proven effective in practice.

A Transformer layer combines self-attention with a skip connection and FFN:

$$\mathbf{X}' = \mathbf{X} + \text{Attn}(\mathbf{X}),$$
$$\mathbf{X}'' = \text{FFN}(\mathbf{X}') := \text{ReLU}(\mathbf{X}'W_1)W_2.$$

Stacking multiple layers forms a Transformer model, resulting in node-level representations. However, due to self-attention's permutation equivariance, Transformers produce identical representations for nodes with matching attributes, regardless of their graph context. Thus, incorporating structural information, typically through positional or structural encoding such as Laplacian positional encoding or random walk structural encoding (Dwivedi et al., 2021; Rampášek et al., 2022), is crucial.

### A.3 State Space Models

As we treat random walks explicitly as sequences, recent advances in long sequence modeling could be leveraged directly to model random walks. SSMs are a type of these models that have shown promising performance in long sequence modeling. SSMs map input sequence $x(t) \in \mathbb{R}$ to some response sequence $y(t) \in \mathbb{R}$ through an implicit state $h(t) \in \mathbb{R}^N$ and three parameters $(A, B, C)$:

$$h'(t) = Ah(t) + Bx(t), \qquad y(t) = Ch(t).$$

For computational reasons, structured SSMs (S4) (Gu et al., 2021) proposes to discretize the above system by introducing a time step variable $\Delta$ and a discretization rule, leading to a reparametrization of the parameters $A$ and $B$. Then, the discrete-time SSMs can be computed in two ways either as a linear recurrence or a global convolution. Recently, a selection mechanism (Gu & Dao, 2023) has been introduced to control which part of the sequence can flow into the hidden states, making the parameters in SSMs time and data-dependent. The proposed model, named Mamba, significantly outperforms its predecessors and results in several successful applications in many tasks. More recently, a bidirectional version of Mamba (Zhu et al., 2024) has been proposed to handle image data, by averaging the representations of both forward and backward sequences after each Mamba block.

## B Additional Remarks on Neural Walker

### B.1 Illustration of the Position Encodings for Random Walks

Here, we give a visual example of the positional encodings that we presented in Section 3.2. The example is shown in Figure 4.

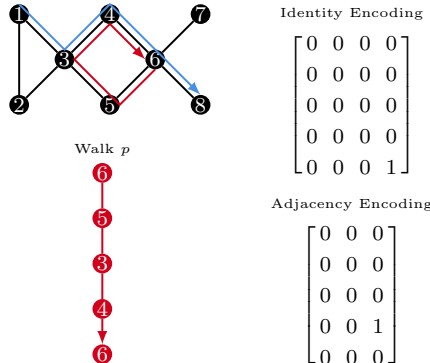

Figure 4: An example of the identity encoding and adjacency encoding presented in Secion 3.2. On the random walk colored in red, we have $\mathrm{id}_W[4,3] = 1$ as $w_4 = w_0 = 6$. We have $\mathrm{adj}_W[3,2] = 1$ as $w_3 w_0 \in E$ is an edge of the graph.

### B.2 Global Message Passing Techniques

Even though long random walks could be sufficient to capture global information, we empirically found that global message passing is still useful in certain tasks. Here, we consider two techniques, namely virtual node and transformer layer. Similar to Gilmer et al. (2017); Tönshoff et al. (2023b), a virtual node layer could be a simple solution to achieve this. Such a layer is explicitly defined as the following:

$$h_V^t(\star) = \mathrm{MLP}\left(h_V^{t-1}(\star) + \sum_{v \in V} h_V^{\mathrm{mp}}(v)\right), \qquad h_V^{\mathrm{vn}}(v) := h_V^{\mathrm{mp}}(v) + h_V^t(\star), \qquad (8)$$

where MLP is a trainable MLP, $h_V^t(\star)$ represents the virtual node embedding at block $t$ and $h_V^0(\star) = 0$. Alternatively, one could use any transformer layer to achieve this. The vanilla transformer layer is given by:

$$h_V^{\mathrm{attn}}(v) = h_V^{\mathrm{mp}}(v) + \mathrm{Attn}(h_V^{\mathrm{mp}})(v), \qquad h_V^{\mathrm{trans}}(v) = h_V^{\mathrm{attn}}(v) + \mathrm{MLP}(h_V^{\mathrm{attn}}(v)), \qquad (9)$$

where $\mathrm{Attn}$ is a trainable scaled dot-product attention layer (Vaswani et al., 2017). This layer is widely used in recent GT models (Ying et al., 2021; Chen et al., 2022a; Rampášek et al., 2022).

### B.3 LIMITATIONS

NeuralWalker demonstrates good scalability to large graphs. However, one potential limitation lies in the trade-off between the sampling efficiency of random walks and graph coverage for very large graphs. In this work, we explored a computationally efficient sampling strategy but probably not with the optimal graph coverage. Investigating more efficient random walk sampling strategies that improve coverage while maintaining computational efficiency could further enhance NeuralWalker's performance.

Additionally, while our experiments focused on modern SSMs like Mamba and S4, which provide efficient implementations, we acknowledge the potential oversight of classical Recurrent Neural Networks (RNNs) such as LSTMs. This consideration is particularly relevant for scenarios with unbounded walk lengths, where RNN-based NeuralWalker models might achieve greater expressivity than their SSM-based counterparts, given RNNs' superior expressivity in terms of circuit complexity when processing unbounded sequences (Merrill et al., 2024).

Finally, we identify a scarcity of publicly available graph datasets with well-defined long-range dependencies. While datasets like LRGB provide valuable examples, the limited number of such datasets hinders comprehensive evaluation and the potential to push the boundaries of long-range dependency capture in graph learning tasks. Furthermore, based on our experiments and Tönshoff et al. (2023a), only 2 out of the 5 datasets in LRGB seem to present long-range dependencies.

### B.4 BROADER IMPACTS

While our research primarily focuses on general graph representation learning, we recognize the importance of responsible and ethical application in specialized fields. When utilized in domains such as drug discovery or computational biology, careful attention must be paid to ensuring the trustworthiness and appropriate use of our method to mitigate potential misuse. Our extensive experiments demonstrate the significant potential of our approach in both social network and biological network analysis, highlighting the promising societal benefits our work may offer in these specific areas.

## C THEORETICAL RESULTS

In this section, we present the background of random walks on graphs and the theoretical properties of NeuralWalker.

**Definition C.1** (Walk feature vector). *For any graph $G = (V, E, x, z)$ and $W \in \mathcal{W}_\ell(G)$, the walk feature vector $X_W$ of $W$ is defined, by concatenating the node and edge feature vectors as well as the positional encodings along $W$ of window size $s = \ell$, as*

$$X_W = (x(w_i), z(w_i w_{i+1}), h_{pe}[i])_{i=0,\ldots,\ell} \in \mathbb{R}^{(\ell+1) \times d_{walk}},$$

*where $h_{pe}$ is the positional encoding in Section 3.2, $\mathbf{z}(w_\ell w_{\ell+1}) = 0$, and $d_{walk} := d + d' + d_{pe}$. By abuse of notation, we denote by $\mathcal{W}(G)$ the set of walk feature vectors on $G$, and by $P(\mathcal{W}(G))$ a distribution of walk feature vectors on $G$.*

**Lemma C.2.** *The walk feature vector with full graph coverage uniquely determines the graph, i.e., for two graphs $G$ and $G'$ in $\mathcal{G}_n$ if there exists a walk $W \in \mathcal{W}_\ell(G)$ visiting all nodes on $G$ and a walk $W' \in \mathcal{W}_\ell$ visiting all nodes on $G'$ such that $X_W = X_{W'}$, then $G$ and $G'$ are isomorphic.*

*Proof.* The proof is immediate following the Observation 1 of Tönshoff et al. (2023b). □

Now if we replace the normalization factor $N_v(P_m(\mathcal{W}_\ell(G)))$ in the walk aggregator in Section 3.3.3 with a simpler deterministic constant $m\ell/|V|$ and apply an average pooling followed by a linear layer $x \mapsto u^\top x + b \in \mathbb{R}$ to the output of the walk aggregator, then the resulting function $g_{f,m,\ell} : \mathcal{G} \to \mathbb{R}$ defined on the graph space $\mathcal{G}$ can be rewritten as the average of some function of walk feature vectors:

$$g_{f,m,\ell}(G) = \frac{1}{m} \sum_{W \in P_m(\mathcal{W}_\ell(G))} f(X_W), \tag{10}$$

where

$$f(X_W) = \frac{1}{\ell} \sum_{w_i \in W} (u^\top f_{seq}(h_W)[i] + b), \tag{11}$$

and $h_W$ defined in Eq. (1) depend on $X_W$.

Note that the above replacement of the normalization factor is not a strong assumption. It is based on the following lemmas:

**Lemma C.3** ((Lovász, 1993)). *Let $G$ be a connected graph. For a random walk $W \sim P(\mathcal{W}(G))$ with $W = (w_0, w_1, \ldots, w_t, \ldots)$, we denote by $P_t$ the distribution of $w_t$. Then,*

$$\pi(v) = \frac{d(v)}{2|E|},$$

*where $d(v)$ denotes the degree of node $v$, is the (unique) stationary distribution, i.e., if $P_0 = \pi$ then $P_t = P_0$ for any $t$. If $P_0 = \pi(v)$, then we have*

$$\mathbb{E}[N_v(P_m(\mathcal{W}_\ell(G)))] = \frac{m\ell d(v)}{2|E|}.$$

*In particular, if $G$ is a regular graph, $\pi(v) = {}^1\!/{}_{|V|}$ is the uniform distribution nad $\mathbb{E}[N_v(P_m(\mathcal{W}_\ell(G)))] = {}^{m\ell}\!/{}_{|V|}$.*

**Lemma C.4** ((Lovász, 1993)). *If $G$ is a non-bipartite graph, then $P_t \to \pi(v)$ as $t \to \infty$.*

The above two lemmas link the random normalization factor to the deterministic one.

If we have a sufficiently large number of random walks, by the law of large numbers, we have

$$g_{f,m,\ell}(G) \xrightarrow{a.s.} g_{f,\ell} := \mathbb{E}_{X_W \sim P(\mathcal{W}_\ell(G))}[f(X_W)], \tag{12}$$

where $\xrightarrow{a.s.}$ denotes the almost sure convergence. This observation inspires us to consider the following integral probability metric (Müller, 1997) comparing distributions of walk feature vectors:

$$d_{\mathcal{F},\ell}(G, G') := \sup_{f \in \mathcal{F}} \left| \mathbb{E}_{X_W \sim P(\mathcal{W}_\ell(G))}[f(X_W)] - \mathbb{E}_{X_{W'} \sim P(\mathcal{W}_\ell(G'))}[f(X_{W'})] \right|, \tag{13}$$

where $\mathcal{F}$ is some functional class, such as the class of neural networks defined by the NeuralWalker model. The following result provides us insight into the rate of convergence of $g_{f,m,\ell}$ to $g_{f,\ell}$:

**Theorem C.5** (Convergence rate). *Assume that $\mathrm{Var}[f(X_W)] = \sigma^2 < \infty$. Then, as $m$ tends to infinity, we have*

$$\sqrt{m} \left( g_{f,m,\ell}(G) - g_{f,\ell}(G) \right) \xrightarrow{d} \mathcal{N}(0, \sigma^2),$$

*where $\xrightarrow{d}$ denotes the convergence in distribution.*

*Proof.* The proof follows the central limit theorem (Dudley, 2018). $\qquad\qquad\square$

$d_{\mathcal{F},\ell}$ is actually a metric on the graph space $\mathcal{G}_n$ of bounded order $n$ if $\mathcal{F}$ is a universal space and $\ell$ is sufficiently large:

**Theorem C.6.** *If $\mathcal{F}$ is a universal space and $\ell \geq 4n^3$, then $d_{\mathcal{F},\ell} : \mathcal{G} \times \mathcal{G} \to \mathbb{R}_+$ is a metric on $\mathcal{G}_n$ satisfying:*

- *(positivity) if $G$ and $G'$ are non-isomorphic, then $d_{\mathcal{F},\ell}(G, G') > 0$.*

- *(symmetry) $d_{\mathcal{F},\ell}(G, G') = d_{\mathcal{F},\ell}(G', G)$.*

- *(triangle inequality) $d_{\mathcal{F},\ell}(G, G'') \leq d_{\mathcal{F},\ell}(G, G') + d_{\mathcal{F},\ell}(G', G'')$.*

*Proof.* The symmetry and triangle inequality are trivial by definition of $d_{\mathcal{F},\ell}$. Let us focus on the positivity. We assume that $d_{\mathcal{F},\ell}(G, G') = 0$. By the universality of $\mathcal{F}$, for any $\varepsilon > 0$ and $f \in C(\mathbb{R}^{d_{\mathrm{walk}}})$, the space of bounded continuous functions on $\mathbb{R}^{d_{\mathrm{walk}}}$, there exists a $g \in \mathcal{F}$ such that

$$\|f - g\|_\infty \leq \varepsilon.$$

We then make the expansion

$$\left|\mathbb{E}_{X_W \sim P(\mathcal{W}_\ell(G))}[f(X_W)] - \mathbb{E}_{X_{W'} \sim P(\mathcal{W}_\ell(G'))}[f(X_{W'})]\right| \leq$$
$$\left|\mathbb{E}_{X_W \sim P(\mathcal{W}_\ell(G))}[f(X_W)] - \mathbb{E}_{X_W \sim P(\mathcal{W}_\ell(G))}[g(X_W)]\right| +$$
$$\left|\mathbb{E}_{X_W \sim P(\mathcal{W}_\ell(G))}[g(X_W)] - \mathbb{E}_{X_{W'} \sim P(\mathcal{W}_\ell(G'))}[g(X_{W'})]\right| +$$
$$\left|\mathbb{E}_{X_{W'} \sim P(\mathcal{W}_\ell(G'))}[g(X_{W'})] - \mathbb{E}_{X_{W'} \sim P(\mathcal{W}_\ell(G'))}[f(X_{W'})]\right|.$$

The first and third terms satisfy

$$\left|\mathbb{E}_{X_W \sim P(\mathcal{W}_\ell(G))}[f(X_W)] - \mathbb{E}_{X_W \sim P(\mathcal{W}_\ell(G))}[g(X_W)]\right| \leq \mathbb{E}_{X_W \sim P(\mathcal{W}_\ell(G))} |f(X_W) - g(X_W)| \leq \varepsilon,$$

and the second term equals 0 by assumption. Hence,

$$\left|\mathbb{E}_{X_W \sim P(\mathcal{W}_\ell(G))}[f(X_W)] - \mathbb{E}_{X_{W'} \sim P(\mathcal{W}_\ell(G'))}[f(X_{W'})]\right| \leq 2\varepsilon,$$

for all $f \in C(\mathbb{R}^{d_{\text{walk}}})$ and $\varepsilon > 0$. This implies $P(\mathcal{W}_\ell(G)) = P(\mathcal{W}_\ell(G'))$ by Lemma 9.3.2 of Dudley (2018), meaning that the distribution of walk feature vectors of length $\ell$ in $G$ is identical to the distribution in $G'$. Without loss of generality, we assume that $G$ and $G'$ are connected and our arguments can be easily generalized to each connected component if $G$ is not connected. Now for a random walk $W \sim P(\mathcal{W}(G))$, let us denote by $T_W$ the number of steps to reach every node on the graph. Then $\mathbb{E}[T_W]$ is called the cover time. A well-known result in graph theory (Aleliunas et al., 1979) states that the cover time is upper bounded:

$$\mathbb{E}[T_W] \leq 4|V||E|.$$

Therefore the cover time for graphs in $\mathcal{G}_n$ is uniformly bounded by $\mathbb{E}[T_W] \leq 4n^3$ as $|V| \leq n$ and $|E| \leq n^2$. Then, by applying Markov's inequality, we have

$$\mathbb{P}[T_W < 4n^3 + \epsilon] = 1 - \mathbb{P}[T_W \geq 4n^3 + \epsilon] \geq 1 - \frac{\mathbb{E}[T_W]}{4n^3 + \epsilon} \geq \frac{\epsilon}{4n^3 + \epsilon} > 0,$$

for any $\epsilon > 0$. Thus, $\mathbb{P}[T_W \leq 4n^3] > 0$ which means that there exists a random walk of not greater than $4n^3$ that visits all nodes in $G$. As a result, there exists a random walk of length $\ell$ reaching all nodes for $\ell \geq 4n^3$. $P(\mathcal{W}_\ell(G)) = P(\mathcal{W}_\ell(G'))$ implies that there also exists a random walk $W'$ in $G'$ such that $X_W = X_{W'}$. As a consequence, $G$ and $G'$ are isomorphic following Lemma C.2. $\quad\square$

Now if we remove the condition on the random walk length $\ell$, we still have a pseudometric space without the positivity in Thm C.6. Moreover, we define the following isomorphism test:

**Definition C.7** ($k$-subgraph isomorphism test)**.** *We define that two graphs $G$ and $G'$ are not distinguishable by the $k$-subgraph isomorphism test iff they have the same set of induced subgraphs of size $k$, i.e., $\mathcal{S}_k(G) = \mathcal{S}_k(G')$ with $\mathcal{S}_k(G)$ denoting the set of induced subgraphs of size $k$.*

And we have the following result which provides a weak positivity of $d_{\mathcal{F},\ell}$ for any $\ell > 0$:

**Theorem C.8.** *If $G$ and $G'$ are distinguishable by the $(\lfloor \ell/2 \rfloor + 1)$-subgraph isomorphism test, then $d_{\mathcal{F},\ell}(G, G') > 0$.*

*Proof.* We assume that $d_{\mathcal{F},\ell}(G, G') = 0$. Using the same arguments as in Thm. C.6, we have $P(\mathcal{W}_\ell(G)) = P(\mathcal{W}_\ell(G'))$. Let $k := \lfloor \ell/2 \rfloor + 1$. For any induced subgraph $H \in \mathcal{S}_k(G)$, there exists a walk of length $\ell$, in the worst case, that visits all its nodes. To see this, let us assume that $G$ is connected without loss of generality. Then, there exists a spanning tree of $H$. Through a depth-first search on this spanning tree, there exists a walk of length $2(k-1) \leq \ell$ that visits all the nodes, by visiting each edge at most twice in the spanning tree. Now as $G$ and $G'$ have the same distributions of walk feature vectors, the same walk feature vector should be found in $G'$, thus $H \in \mathcal{S}_k(G')$. Thus, we have $\mathcal{S}_k(G) \subseteq \mathcal{S}_k(G')$. Similarly, we have the other inclusion and therefore $\mathcal{S}_k(G) = \mathcal{S}_k(G')$. $\quad\square$

## C.1 STABILITY RESULTS

Now that we have a metric space $(\mathcal{G}_n, d_{\mathcal{F},\ell})$ with $\ell \geq 4n^3$, we can show some useful properties of $g_{f,\ell}$:

**Theorem C.9** (Lipschiz continuity of $g_{f,\ell}$). *For any $G$ and $G'$ in $\mathcal{G}_n$, if $\mathcal{F}$ is a functional space containing $f$, we have*

$$|g_{f,\ell}(G) - g_{f,\ell}(G')| \leq d_{\mathcal{F},\ell}(G, G'). \tag{14}$$

*Proof.* The proof is immediate from the definition of $d_{\mathcal{F},\ell}$. □

The Lipschiz property is needed for stability to perturbations in the sense that if $G'$ is close to $G$ in $(\mathcal{G}_n, d_{\mathcal{F},\ell})$, then their images by $g_{f,\ell}$ (output of the model) are also close.

### C.2  EXPRESSIVITY RESULTS

**Theorem C.10** (Injectivity of $g_{f,\ell}$). *Assume $\mathcal{F}$ is a universal space. If $G$ and $G'$ are non-isomorphic graphs, then there exists a $f \in \mathcal{F}$ such that $g_{f,\ell}(G) \neq g_{f,\ell}(G')$ if $\ell \geq 4\max\{|V|, |V'|\}^3$.*

*Proof.* We can prove this by contrapositive. We note that $G, G' \in \mathcal{G}_{n_{\max}}$ with $n_{\max} := \max\{|V|, |V'|\}$. Assume that for all $f \in \mathcal{F}$, $g_{f,\ell}(G) = g_{f,\ell}(G')$. This implies that $d_{\mathcal{F},\ell}(G, G') = 0$. Then by the positivity of $d_{\mathcal{F},\ell}$ in $\mathcal{G}_{n_{\max}}$, $G$ and $G'$ are isomorphic. □

The injectivity property ensures that our model with a sufficiently large number of sufficiently long ($\geq 4n^3$) random walks can distinguish between non-isomorphic graphs, highlighting its expressive power.

Complementary to the above results, we now show that the expressive power of our model exceeds that of ordinary message passing neural networks even when considering random walks of small size. Additionally, we show that the expressive power of our model is stronger than the subgraph isomorphism test up to a certain size.

We first state a result on the expressive power of the walk aggregator function (defined in Sect. 3.3.3). We show that there exist walk aggregator functions such that for a node $v$, this function encodes a (multi-)set of induced subgraphs that $v$ is part of. Let $G_v$ denote the (isomorphism class of) graph $G$ rooted at node $v$ and let the set $x_\ell(G, v) = \{\{G_v = G[\{w_0, \ldots, w_k = v\}], W = (w_0, \ldots, w_\ell), \ell \geq k, W \in \mathcal{W}_\ell(G)\}\}$.

**Lemma C.11.** *There exists a walk aggregator function $f_{agg}$ such that for any nodes $v \in G$, $v' \in G'$ and walk length $\ell$, it holds that $f_{agg}(v) = f_{agg}(v')$ if and only if $x_\ell(G, v) = x_\ell(G', v')$.*

*Proof.* Recall from Sect. 3.2 that the positional encoding of a walk $W$ encodes the pairwise adjacency of nodes contained in $W$. In fact, this encoding also contains the complete information on subgraphs induced by prefixes of $W$. More formally, for a length-$\ell$ walk $W \in \mathcal{W}_\ell(G)$, the first $k + 1$ rows of the corresponding walk feature vector $X_W$ encode the induced subgraph $G[\{w_0, \ldots, w_k\}]$. With $w_k = v$, it can also be inferred about the structural role of $v$ in $G[\{w_0, \ldots, w_k\}]$. Further recall that walk aggregator functions aggregate information from all walks passing through a node. Thus, we can design a function $f_{agg}$ that aggregates the induced subgraph information for sets of subgraphs into node embeddings. More precisely, for a node $v \in G$ and the set of walks $W_\ell(G)$, there exists such a function $f_{agg}(v)$ that maps $v$ to an embedding which encodes the set $x_\ell(G, v)$. By considering the complete set of walks $W_\ell(G)$, we guarantee a deterministic embedding. Assuming a sufficiently powerful neural network, it can be shown that such a function $f_{agg}$ can be realized by our model. □

We are ready to prove results on NeuralWalker's ability to distinguish between substructures.

**Theorem C.12.** *For any $\ell \geq 2$, NeuralWalker equipped with the complete walk set $\mathcal{W}_\ell$ is strictly more expressive than 1-WL and the $(\lfloor \ell/2 \rfloor + 1)$-subgraph isomorphism test, and thus ordinary MPNNs.*

*Proof.* For the subgraph isomorphism test, we simply use the above theorem and Thm. C.8 which suggests that there exists a $f \in \mathcal{F}$ such that $g_{f,\ell}(G) \neq g_{f,\ell}(G')$ if $G$ and $G'$ are distinguishable by the $(\lfloor \ell/2 \rfloor + 1)$-subgraph isomorphism test.

Concerning the comparison to the 1-WL test, we utilize the result of lemma C.11. A walk aggregator function $f_{agg}$, as defined in this lemma, maps nodes $v \in G$ to an embedding encoding the subgraph information $x_\ell(G, v)$. For instance, using walk length $\ell = 2$, the node embeddings contain

Table 7: Summary of the datasets Dwivedi et al. (2023; 2022); Hu et al. (2020a) used in this study.

| DATASET | # GRAPHS | AVG. # NODES | AVG. # EDGES | DIRECTED | PREDICTION LEVEL | PREDICTION TASK | METRIC |
|---------|----------|--------------|--------------|----------|------------------|-----------------|--------|
| ZINC | 12,000 | 23.2 | 24.9 | No | GRAPH | REGRESSION | MEAN ABS. ERROR |
| MNIST | 70,000 | 70.6 | 564.5 | YES | GRAPH | 10-CLASS CLASSIF. | ACCURACY |
| CIFAR10 | 60,000 | 117.6 | 941.1 | YES | GRAPH | 10-CLASS CLASSIF. | ACCURACY |
| PATTERN | 14,000 | 118.9 | 3,039.3 | No | INDUCTIVE NODE | BINARY CLASSIF. | ACCURACY |
| CLUSTER | 12,000 | 117.2 | 2,150.9 | No | INDUCTIVE NODE | 6-CLASS CLASSIF. | ACCURACY |
| PASCALVOC-SP | 11,355 | 479.4 | 2,710.5 | No | INDUCTIVE NODE | 21-CLASS CLASSIF. | F1 SCORE |
| COCO-SP | 123,286 | 476.9 | 2,693.7 | No | INDUCTIVE NODE | 81-CLASS CLASSIF. | F1 SCORE |
| PEPTIDES-FUNC | 15,535 | 150.9 | 307.3 | No | GRAPH | 10-TASK CLASSIF. | AVG. PRECISION |
| PCQM-CONTACT | 529,434 | 30.1 | 61.0 | No | INDUCTIVE LINK | LINK RANKING | MRR |
| PEPTIDES-STRUCT | 15,535 | 150.9 | 307.3 | No | GRAPH | 11-TASK REGRESSION | MEAN ABS. ERROR |
| OGBG-MOLPCBA | 437,929 | 26.0 | 28.1 | No | GRAPH | 128-TASK CLASSIF. | AVG. PRECISION |
| OGBG-PPA | 158,100 | 243.4 | 2,266.1 | No | GRAPH | 37-TASK CLASSIF. | ACCURACY |
| OGBG-CODE2 | 452,741 | 125.2 | 124.2 | YES | GRAPH | 5 TOKEN SEQUENCE | F1 SCORE |

information about the (multi-)set of triangles that nodes are part of. Then, using NeuralWalker's message passing layers, this subgraph information leads to more powerful graph representations when compared to ordinary message passing without such enriched node embeddings. In fact, analogously to e.g. Bouritsas et al. (2023), it can be shown that with a sufficient number of message passing layers and a powerful readout network, the resulting graph representations are strictly more powerful than ordinary MPNNs.

$\square$

### C.3 COMPLEXITY RESULTS

**Theorem C.13** (Complexity). *The complexity of NeuralWalker, when used with the Mamba sequence layer (Gu & Dao, 2023), is $O(kdn(\gamma\ell + \beta))$, where $k, d, n, \gamma, \ell, \beta$ denote the number of layers, hidden dimensions, the (maximum) number of nodes, sampling rate, length of random walks, and the average degree, respectively.*

*Proof.* The complexity of sampling random walks is $O(n\gamma\ell)$. The Mamba model with $k$ layers and hidden dimensions $d$ operates on $O(n\gamma)$ random walks of length $\ell$. As Mamba scales linearly to the sequence length, number of layers, and hidden dimensions (Gu & Dao, 2023), its complexity is $O(kdn\gamma\ell)$. The complexity of $k$ message passing layers of hidden dimensions $d$ is $O(kdn\beta)$ where $\beta$ should be much smaller than $\gamma\ell$ in general. $\square$

It is worth noting that the space complexity of NeuralWalker is the same as its time complexity, due to the fact that Mamba is linear to the sequence length for both time and space complexity (Gu & Dao, 2023).

## D EXPERIMENTAL DETAILS AND ADDITIONAL RESULTS

In this section, we provide implementation details and additional experimental results

### D.1 DATASET DESCRIPTION

We provide details of the datasets used in our experiments. For each dataset, we follow their respective training protocols and use the standard train/validation/test splits and evaluation metrics.

**ZINC (MIT License) (Dwivedi et al., 2023).** The ZINC dataset is a subset of the ZINC database, containing 12,000 molecular graphs representing commercially available chemical compounds. These graphs range from 9 to 37 nodes in size, with each node corresponding to a "heavy atom" (one of 28 possible types) and each edge representing a bond (one of 3 types). The goal is to predict the constrained solubility (logP) using regression. The dataset is conveniently pre-split for training, validation, and testing, with a standard split of 10,000/1,000/1,000 molecules for each set, respectively.

Table 8: Summary of the datasets for transductive node classification Platonov et al. (2022); Leskovec & Krevl (2014) used in this study.

| DATASET | HOMOPHILY SCORE | # NODES | # EDGES | # CLASSES | METRIC |
|---|---|---|---|---|---|
| ROMAN-EMPIRE | 0.023 | 22,662 | 32,927 | 18 | ACCURACY |
| AMAZON-RATINGS | 0.127 | 24,492 | 93,050 | 5 | ACCURACY |
| MINESWEEPER | 0.009 | 10,000 | 39,402 | 2 | ROC AUC |
| TOLOKERS | 0.187 | 11,758 | 519,000 | 2 | ROC AUC |
| QUESTIONS | 0.072 | 48,921 | 153,540 | 2 | ROC AUC |
| POKEC | 0.000 | 1,632,803 | 30,622,564 | 2 | ACCURACY |

**MNIST and CIFAR10 (CC BY-SA 3.0 and MIT License) Dwivedi et al. (2023).** MNIST and CIFAR10 are adapted for graph-based learning by converting each image into a graph. This is achieved by segmenting the image into superpixels using SLIC (Simple Linear Iterative Clustering) and then connecting each superpixel to its 8 nearest neighbors. The resulting graphs maintain the original 10-class classification task and standard dataset splits (*i.e.,* 55K/5K/10K train/validation/test for MNIST and 45K/5K/10K for CIFAR10.).

**PATTERN and CLUSTER (MIT License) (Dwivedi et al., 2023).** PATTERN and CLUSTER are synthetic graph datasets constructed using the Stochastic Block Model (SBM). They offer a unique challenge for inductive node-level classification, where the goal is to predict the class label of unseen nodes. PATTERN: This dataset presents the task of identifying pre-defined sub-graph patterns (100 possible) embedded within the larger graph. These embedded patterns are generated from distinct SBM parameters compared to the background graph, requiring the model to learn these differentiating connection characteristics. CLUSTER: Each graph in CLUSTER consists of six pre-defined clusters generated using the same SBM distribution. However, only one node per cluster is explicitly labeled with its unique cluster ID. The task is to infer the cluster membership (ID) for all remaining nodes based solely on the graph structure and node connectivity information.

**PASCALVOC-SP and COCO-SP (Custom license for Pascal VOC 2011 respecting Flickr terms of use, and CC BY 4.0 license) (Dwivedi et al., 2022).** PascalVOC-SP and COCO-SP are graph datasets derived from the popular image datasets Pascal VOC and MS COCO, respectively. These datasets leverage SLIC superpixellization, a technique that segments images into regions with similar properties. In both datasets, each superpixel is represented as a node in a graph, and the classification task is to predict the object class that each node belongs to.

**PEPTIDES-FUNC and PEPTIDES-STRUCT (CC BY-NC 4.0) (Dwivedi et al., 2022).** Peptides-func and Peptides-struct offer complementary views of peptide properties by leveraging atomic graphs derived from the SATPdb database. Peptides-func focuses on multi-label graph classification, aiming to predict one or more functional classes (out of 10 non-exclusive categories) for each peptide. In contrast, Peptides-struct employs graph regression to predict 11 continuous 3D structural properties of the peptides.

**PCQM-CONTACT (CC BY 4.0) (Dwivedi et al., 2022).** The PCQM-Contact dataset builds upon PCQM4Mv2 (Hu et al., 2020a) by incorporating 3D molecular structures. This enables the task of binary link prediction, where the goal is to identify pairs of atoms (nodes) that are considered to be in close physical proximity (less than 3.5 angstroms) in 3D space, yet appear far apart (more than 5 hops) when looking solely at the 2D molecular graph structure. The standard evaluation metric for this ranking task is Mean Reciprocal Rank (MRR). As noticed by Tönshoff et al. (2023a), the original implementation by Dwivedi et al. (2022) suffers from false negatives and self-loops. Thus, we use the filtered version of the MRR provided by Tönshoff et al. (2023a).

**OGBG-MOLPCBA (MIT License) (Hu et al., 2020a).** The ogbg-molpcba dataset, incorporated by the Open Graph Benchmark (OGB) (Hu et al., 2020a) from MoleculeNet, focuses on multi-task binary classification of molecular properties. This dataset leverages a standardized node (atom) and edge (bond) feature representation that captures relevant chemophysical information. Derived from PubChem BioAssay, ogbg-molpcba offers the task of predicting the outcome of 128 distinct bioassays, making it valuable for studying the relationship between molecular structure and biological activity.

**OGBG-PPA (CC-0 license) (Hu et al., 2020a).** The PPA dataset, introduced by OGB (Hu et al., 2020a), focuses on species classification. This dataset represents protein-protein interactions within a network, where each node corresponds to a protein and edges denote associations between them. Edge attributes provide additional information about these interactions, such as co-expression levels. We employ the standard dataset splits established by OGB (Hu et al., 2020a) for our analysis.

**OGBG-CODE2 (MIT License) (Hu et al., 2020a).** CODE2 (Hu et al., 2020a) is a dataset containing source code from the Python programming language. It is made up of Abstract Syntax Trees where the task is to classify the sub-tokens that comprise the method name. We use the standard splits provided by OGB (Hu et al., 2020a).

**ROMAN-EMPIRE (MIT License) (Platonov et al., 2022).** This dataset creates a graph from the Roman Empire Wikipedia article. Each word becomes a node, and edges connect words that are either sequential in the text or grammatically dependent (based on the dependency tree). Nodes are labeled by their syntactic role ( 17 most frequent roles are selected as unique classes and all the other roles are grouped into the 18th class). We use the standard splits provided by Platonov et al. (2022).

**AMAZON-RATINGS (MIT License) (Platonov et al., 2022).** Based on the Amazon product co-purchase data, this dataset predicts a product's average rating (5 classes). Products (books, etc.) are nodes, connected if frequently bought together. Mean fastText embeddings are used for product descriptions as node features and focus on the largest connected component for efficiency (5-core). We use the standard splits provided by Platonov et al. (2022).

**MINESWEEPER (MIT License) (Platonov et al., 2022).** This is a synthetic dataset with a regular 100x100 grid where nodes represent cells. Each node connects to its eight neighbors (except edges). 20% of nodes are randomly mined. The task is to predict which are mines. Node features are one-hot-encoded numbers of neighboring mines, but are missing for 50% of nodes (marked by a separate binary feature). This grid structure differs from other datasets due to its regularity (average degree: 7.88). Since mines are random, both adjusted homophily and label informativeness are very low. We use the standard splits provided by Platonov et al. (2022).

**TOLOKERS (MIT License) (Platonov et al., 2022).** This dataset features workers (nodes) from crowdsourcing projects. Edges connect workers who have collaborated on at least one of the 13 projects. The task is to predict banned workers. Node features include profile information and performance statistics. This graph (11.8K nodes, avg. degree 88.28) is significantly denser compared to other datasets. We use the standard splits provided by Platonov et al. (2022).

**QUESTIONS (MIT License) (Platonov et al., 2022).** This dataset focuses on user activity prediction. Users are nodes, connected if they answered each other's questions (Sept 2021 - Aug 2022). The task is to predict which users remained active. User descriptions (if available) are encoded using fastText embeddings. Notably, 15% lack descriptions and are identified by a separate feature. We use the standard splits provided by Platonov et al. (2022).

**POKEC (unknown License) (Leskovec & Krevl, 2014).** This dataset was retrieved from SNAP (Leskovec & Krevl, 2014) and preprocessed by Lim et al. (2021). The dataset contains anonymized data of the whole network of Pokec, the most popular online social network in Slovakia which has been provided for more than 10 years and connects more than 1.6 million people. Profile data contains gender, age, hobbies, interests, education, etc, and the task is to predict the gender. The dataset was not released with a license. Thus, we only provide numerical values without any raw texts from the dataset.

## D.2 COMPUTING DETAILS

We implemented our models using PyTorch Geometric (Fey & Lenssen, 2019) (MIT License). Experiments were conducted on a shared computing cluster with various CPU and GPU configurations, including a mix of NVIDIA A100 (40GB) and H100 (80GB) GPUs. Each experiment was allocated resources on a single GPU, along with 4-8 CPUs and up to 60GB of system RAM. The run-time of each model was measured on a single NVIDIA A100 GPU.

Table 9: Hyperparameters for the 5 datasets from GNN Benchmarks (Dwivedi et al., 2023).

| HYPERPARAMETER | ZINC | MNIST | CIFAR10 | PATTERN | CLUSTER |
|---|---|---|---|---|---|
| # BLOCKS | 3 | 3 | 3 | 3 | 16 |
| HIDDEN DIM | 80 | 80 | 80 | 80 | 32 |
| SEQUENCE LAYER | | | MAMBA (BIDIRECTIONAL) | | |
| LOCAL MESSAGE PASSING | | | GIN | | |
| GLOBAL MESSAGE PASSING | VN | NONE | NONE | VN | VN |
| DROPOUT | 0.0 | 0.0 | 0.0 | 0.0 | 0.0 |
| GRAPH POOLING | SUM | MEAN | MEAN | – | – |
| RW SAMPLING RATE | 1.0 | 0.5 | 0.5 | 0.5 | 0.5 |
| RW LENGTH | 50 | 50 | 50 | 100 | 200 |
| RW POSITION ENCODING WINDOW SIZE | 8 | 8 | 8 | 16 | 32 |
| BATCH SIZE | 50 | 32 | 32 | 32 | 32 |
| LEARNING RATE | 0.002 | 0.002 | 0.002 | 0.002 | 0.01 |
| # EPOCHS | 2000 | 100 | 100 | 100 | 100 |
| # WARMUP EPOCHS | 50 | 5 | 5 | 5 | 5 |
| WEIGHT DECAY | 0.0 | 1E-6 | 1E-6 | 0.0 | 0.0 |
| # PARAMETERS | 502K | 112K | 112K | 504K | 525K |
| TRAINING TIME (EPOCH/TOTAL) | 16S/8.4H | 90S/2.5H | 95S/2.6H | 57S/1.6H | 241S/6.7H |

Table 10: Hyperparameters for the 5 datasets from LRGB (Dwivedi et al., 2022).

| HYPERPARAMETER | PASCALVOC-SP | COCO-SP | PEPTIDES-FUNC | PEPTIDES-STRUCT | PCQM-CONTACT |
|---|---|---|---|---|---|
| # BLOCKS | 6 | 6 | 6 | 6 | 3 |
| HIDDEN DIM | 52 | 56 | 56 | 56 | 80 |
| SEQUENCE LAYER | | | MAMBA (BIDIRECTIONAL) | | |
| LOCAL MESSAGE PASSING | | | GIN | | |
| GLOBAL MESSAGE PASSING | TRANS. | NONE | VN | VN | VN |
| DROPOUT | 0.0 | 0.0 | 0.0 | 0.0 | 0.0 |
| GRAPH POOLING | – | – | MEAN | MEAN | – |
| RW SAMPLING RATE | 0.5 | 0.25 | 0.5 | 0.5 | 0.5 |
| RW LENGTH | 100 | 100 | 100 | 100 | 75 |
| RW POSITION ENCODING WINDOW SIZE | 16 | 16 | 16 | 32 | 16 |
| BATCH SIZE | 32 | 32 | 32 | 32 | 256 |
| LEARNING RATE | 0.002 | 0.002 | 0.002 | 0.004 | 0.001 |
| # EPOCHS | 200 | 200 | 200 | 200 | 150 |
| # WARMUP EPOCHS | 10 | 10 | 10 | 10 | 10 |
| WEIGHT DECAY | 1E-06 | 0.0 | 0.0 | 0.0 | 0.0 |
| # PARAMETERS | 556K | 492K | 530K | 541K | 505K |
| TRAINING TIME (EPOCH/TOTAL) | 218S/12H | 1402S/78H | 112S/6.2H | 112S/6.2H | 528S/22H |

### D.3 HYPERPARAMETERS

Given the large number of hyperparameters and datasets, we did not perform an exhaustive search beyond the ablation studies in Section 5.3. For each dataset, we then adjusted the number of layers, the hidden dimension, the learning rate, the weight decay based on hyperparameters reported in the related literature (Rampášek et al., 2022; Tönshoff et al., 2023b; Deng et al., 2024; Tönshoff et al., 2023a).

For the datasets from Benchmarking GNNs (Dwivedi et al., 2023) and LRGB (Dwivedi et al., 2022), we follow the commonly used parameter budgets of 500K parameters.

For the node classification datasets from Platonov et al. (2022) and Leskovec & Krevl (2014), we strictly follow the experimental setup from the state-of-the-art method Polynormer (Deng et al., 2024). We only replace the global attention blocks from Polynormer with NeuralWalker's walk encoder blocks and use the same hyperparameters selected by Polynormer (Deng et al., 2024).

We use the AdamW optimizer throughout our experiments with the default beta parameters in Pytorch. We use a linear warm-up increase of the learning rate at the beginning of the training followed by its cosine decay as in Rampášek et al. (2022). The test sampling rate is always set to 1.0 if not specified. The detailed hyperparameters used in NeuralWalker as well as the model sizes and runtime on different datasets are provided in Table 9, 10, 11, and 12.

Table 11: Hyperparameters for the 3 datasets from OGB (Hu et al., 2020a).

| HYPERPARAMETER | OGBG-MOLPCBA | OGBG-PPA | OGBG-CODE2 |
|---|---|---|---|
| # BLOCKS | 4 | 1 | 3 |
| HIDDEN DIM | 500 | 384 | 256 |
| SEQUENCE LAYER | CONV. | CONV. | CONV. |
| LOCAL MESSAGE PASSING | GATEDGCN | GIN | GIN |
| GLOBAL MESSAGE PASSING | VN | PERFORMER | TRANS. |
| DROPOUT | 0.4 | 0.4 | 0.0 |
| GRAPH POOLING | MEAN | MEAN | MEAN |
| RW SAMPLING RATE | 0.5 | 0.5 | 0.5 |
| RW LENGTH | 25 | 200 | 100 |
| RW POSITION ENCODING WINDOW SIZE | 8 | 32 | 64 |
| BATCH SIZE | 512 | 32 | 32 |
| LEARNING RATE | 0.002 | 0.002 | 0.0003 |
| # EPOCHS | 100 | 200 | 30 |
| # WARMUP EPOCHS | 5 | 10 | 2 |
| WEIGHT DECAY | 0.0 | 0.0 | 0.0 |
| # PARAMETERS | 13.0M | 3.1M | 12.5M |
| TRAINING TIME (EPOCH/TOTAL) | 226S/6.3H | 671S/37H | 1597S/13.3H |

Table 12: Hyperparameters for node classification datasets from Platonov et al. (2022) and Leskovec & Krevl (2014). The other hyperparameters strictly follow Polynormer (Deng et al., 2024).

| HYPERPARAMETER | ROMAN-EMPIRE | AMAZON-RATINGS | MINESWEEPER | TOLOKERS | QUESTIONS | POKEC |
|---|---|---|---|---|---|---|
| SEQUENCE LAYER | | | MAMBA | | | CONV. |
| DROPOUT | 0.3 | 0.2 | 0.3 | 0.1 | 0.2 | 0.1 |
| RW SAMPLING RATE | 0.01 | 0.01 | 0.01 | 0.01 | 0.01 | 0.001 |
| RW TEST SAMPLING RATE | 0.1 | 0.1 | 0.1 | 0.1 | 0.05 | 0.001 |
| RW LENGTH | 1000 | 1000 | 1000 | 1000 | 1000 | 500 |
| RW POSITION ENCODING WINDOW SIZE | 8 | 8 | 8 | 8 | 8 | 8 |
| LEARNING RATE | 0.0005 | 0.0005 | 0.0005 | 0.001 | 5E-5 | 0.0005 |
| TRAINING TIME (EPOCH/TOTAL) | 0.50S/0.35H | 0.6S/0.45H | 0.22S/0.12H | 0.67S/0.19H | 0.67S/0.32H | 6.44S/4.5H |

## D.4 ADDITIONAL RESULTS FOR ABLATION STUDIES

We provide more detailed results for ablation studies in Table 13. A time comparison of CNN and Mamba used as the sequence layers in NeuralWalker is presented in Table 14. We also report the GPU memory usage for NeuralWalker on the ZINC, CIFAR10, and PascalVOC-SP datasets in Figure 5, analyzing the impact of varying the random walk sampling rate and length hyperparameters. All the other hyperparameters are fixed to the values specified in Table 9 and 10. The figure demonstrates that NeuralWalker scales strictly linearly with both the sampling rate and length, highlighting the model's potential for application to very large datasets.

Table 13: Ablation studies of NeuralWalker on different choices of the sequence layer, local and global message passing. Validation performances with mean ± std of 4 runs are reported. We compare different choices of sequence layers (Mamba, S4, CNN, and Transformer), local (with or without GIN) and global (virtual node (VN), Transformer, or none (w/o)) message passing layers. Note that the row highlighted with the light gray color corresponds to the choices of CRaWL (Tönshoff et al., 2023b).

| SEQUENCE LAYER | LOCAL MP | GLOBAL MP | ZINC | CIFAR10 | PASCALVOC-SP |
|---|---|---|---|---|---|
| MAMBA | GIN | VN | **0.078 ± 0.004** | 78.610 ± 0.524 | 0.4672 ± 0.0077 |
| MAMBA | GIN | TRANS. | 0.083 ± 0.003 | 80.755 ± 0.467 | **0.4877 ± 0.0042** |
| MAMBA | GIN | W/O | 0.085 ± 0.003 | **80.885 ± 0.769** | 0.4611 ± 0.0036 |
| MAMBA | W/O | VN | 0.086 ± 0.008 | 78.025 ± 0.552 | 0.4570 ± 0.0064 |
| MAMBA | W/O | W/O | 0.090 ± 0.002 | 79.035 ± 0.850 | 0.4525 ± 0.0044 |
| MAMBA (W/O BID) | GIN | VN | 0.089 ± 0.004 | 74.910 ± 0.547 | 0.4522 ± 0.0063 |
| S4 | GIN | VN | 0.082 ± 0.004 | 77.970 ± 0.506 | 0.4559 ± 0.0064 |
| CNN | GIN | VN | 0.088 ± 0.004 | 80.240 ± 0.767 | 0.4652 ± 0.0058 |
| CNN | GIN | TRANS. | 0.092 ± 0.004 | 80.665 ± 0.408 | 0.4790 ± 0.0081 |
| CNN | GIN | W/O | 0.102 ± 0.003 | 80.020 ± 0.279 | 0.4155 ± 0.0050 |
| CNN | W/O | W/O | 0.116 ± 0.003 | 78.760 ± 0.242 | 0.3954 ± 0.0080 |
| TRANS. | GIN | VN | 0.084 ± 0.003 | 72.850 ± 0.373 | 0.4316 ± 0.0072 |

Table 14: Training time (Epoch/Total) comparison when using CNN and Mamba as the sequence layer in NeuralWalker. The time values are measured on a single A100 GPU.

| Sequence Layer | ZINC | CIFAR10 | PascalVOC-SP |
|---|---|---|---|
| Mamba | 16s/8.4h | 95s/2.6h | 218s/12h |
| CNN | 8.9s/5h | 29s/0.8h | 71s/3.9h |

Figure 5: GPU memory usage of NeuralWalker for training one iteration on the ZINC, CIFAR10, and PascalVOC-SP datasets. The hyperparameters are fixed to the values specified in Table 9 and 10 when varying the random walk sampling rate and length. Memory consumption is measured using the `Pytorch.profiler` library.

## D.5 Detailed Results and Robustness to Sampling Variability

Since NeuralWalker's output depends on the sampled random walks, we evaluate its robustness to sampling variability. Following Tönshoff et al. (2023b), we measure the local standard deviation (local std) by computing the standard deviation of performance metrics obtained with five independent sets of random walks (details in Tönshoff et al. (2023b)). The complete results for all datasets are presented in Table 15. Notably, by comparing the local std to the cross-model std obtained from training different models with varying random seeds, we consistently observe a smaller local std. This finding suggests that NeuralWalker's predictions are robust to the randomness inherent in the random walk sampling process.

## D.6 Additional Results on Training and Inference Time

In order to provide a comprehensive insight into the scalability of our model across all datasets, we report in Table 16 statistics for the number and length of random walks selected for each dataset, along with the corresponding training and inference times. The results demonstrate that random walk sampling is highly efficient, and typically faster than model training or inference. Additionally, our model exhibits reasonable scalability with increasing graph size.

Table 15: Detailed results for all the datasets. Note that different metrics are used to measure the performance on the datasets. For each experiment, we provide the cross-model std using different random seeds and the local std using different sets of random walks.

| DATASET | METRIC | TEST | | | VALIDATION | |
| --- | --- | --- | --- | --- | --- | --- |
| | | SCORE | CROSS MODEL STD | LOCAL STD | SCORE | CROSS-MODEL STD |
| **ZINC** | MAE | 0.0646 | 0.0007 | 0.0005 | 0.0782 | 0.0038 |
| **MNIST** | ACC | 0.9876 | 0.0008 | 0.0003 | 0.9902 | 0.0006 |
| **CIFAR10** | ACC | 0.8003 | 0.0019 | 0.0009 | 0.8125 | 0.0053 |
| **PATTERN** | ACC | 0.8698 | 0.0001 | 0.0001 | 0.8689 | 0.0003 |
| **CLUSTER** | ACC | 0.7819 | 0.0019 | 0.0004 | 0.7827 | 0.0007 |
| **PASCALVOC-SP** | F1 | 0.4912 | 0.0042 | 0.0019 | 0.5053 | 0.0084 |
| **COCO-SP** | F1 | 0.4398 | 0.0033 | 0.0011 | 0.4446 | 0.0030 |
| **PEPTIDES-FUNC** | AP | 0.7096 | 0.0078 | 0.0014 | 0.7145 | 0.0033 |
| **PEPTIDES-STRUCT** | AP | 0.2463 | 0.0005 | 0.0004 | 0.2389 | 0.0021 |
| **PCQM-CONTACT** | MRR | 0.4707 | 0.0007 | 0.0002 | 0.4743 | 0.0006 |
| **OGBG-MOLPCBA** | AP | 0.3086 | 0.0031 | 0.0010 | 0.3160 | 0.0032 |
| **OGBG-PPA** | ACC | 0.7888 | 0.0059 | 0.0004 | 0.7460 | 0.0058 |
| **OGBG-CODE2** | F1 | 0.1957 | 0.0025 | 0.0005 | 0.1796 | 0.0031 |
| **ROMAN-EMPIRE** | ACC | 0.9292 | 0.0036 | 0.0005 | 0.9310 | 0.0032 |
| **AMAZON-RATINGS** | ACC | 0.5458 | 0.0036 | 0.0009 | 0.5491 | 0.0049 |
| **MINESWEEPER** | ROC AUC | 0.9782 | 0.0040 | 0.0003 | 0.9794 | 0.0047 |
| **TOLOKERS** | ROC AUC | 0.8556 | 0.0075 | 0.0010 | 0.8540 | 0.0096 |
| **QUESTIONS** | ROC AUC | 0.7852 | 0.0113 | 0.0009 | 0.7902 | 0.0086 |
| **POKEC** | ACC | 0.8646 | 0.0009 | 0.0001 | 0.8644 | 0.0003 |

| | | | | | Training Time (s) | | Inference Time (s) | |
| --- | --- | --- | --- | --- | --- | --- | --- | --- |
| Dataset | # Nodes ($n$) | # RWs ($n\gamma$) | RW length ($\ell$) | $\gamma\ell$ | Model | RW Sampling | Model | RW Sampling |
| ZINC | 23.2 | 23.2 | 50 | 50.0 | 0.001011 | 1.3e-05 | 0.000319 | 0.000132 |
| OGBG-MOLPCBA | 26.0 | 13.0 | 25 | 12.5 | 0.000373 | 4e-06 | 0.000222 | 1.1e-05 |
| PCQM-Contact | 30.1 | 15.05 | 75 | 37.5 | 0.000725 | 4e-06 | 0.001792 | 3e-05 |
| MNIST | 70.6 | 35.3 | 50 | 25.0 | 0.001063 | 1.4e-05 | 0.000497 | 3.2e-05 |
| CLUSTER | 117.2 | 58.6 | 200 | 100.0 | 0.014039 | 0.000133 | 0.006935 | 0.001245 |
| CIFAR10 | 117.6 | 58.8 | 50 | 25.0 | 0.001273 | 1e-05 | 0.000631 | 6.2e-05 |
| PATTERN | 118.9 | 59.45 | 100 | 50.0 | 0.002797 | 7.9e-05 | 0.001502 | 0.00025 |
| OGBG-CODE2 | 125.2 | 62.6 | 100 | 50.0 | 0.002597 | 5.7e-05 | 0.001586 | 0.002108 |
| Peptides-Func | 150.9 | 75.45 | 100 | 50.0 | 0.00609 | 6.4e-05 | 0.002655 | 0.000149 |
| Peptides-Struct | 150.9 | 75.45 | 100 | 50.0 | 0.006162 | 0.000104 | 0.002695 | 0.00028 |
| OGBG-PPA | 243.4 | 121.7 | 200 | 100.0 | 0.003708 | 0.000615 | 0.002411 | 0.004862 |
| COCO-SP | 476.9 | 119.225 | 100 | 25.0 | 0.008137 | 1.7e-05 | 0.007638 | 0.00031 |
| PASCALVOC-SP | 479.4 | 239.7 | 100 | 50.0 | 0.015187 | 0.000274 | 0.007574 | 0.000936 |
| MINESWEEPER | 10000.0 | 100.0 | 1000 | 10.0 | 0.15 | 0.024003 | 0.016 | 0.160348 |
| TOLOKERS | 11758.0 | 117.58 | 1000 | 10.0 | 0.31 | 0.33813 | 0.012 | 2.342794 |
| ROMAN-EMPIRE | 22662.0 | 226.62 | 1000 | 10.0 | 0.31 | 0.037853 | 0.017 | 0.316381 |
| AMAZON-RATINGS | 24492.0 | 244.92 | 1000 | 10.0 | 0.14 | 0.040961 | 0.016 | 0.301582 |
| QUESTIONS | 48921.0 | 489.21 | 1000 | 10.0 | 0.26 | 0.400535 | 0.01 | 1.841745 |
| POKEC | 1632803.0 | 1632.803 | 500 | 0.5 | 2.1 | 3.177021 | 102.925 | 9.621128 |

Table 16: Statistics for the number and length of random walks selected for each dataset, along with the corresponding training and inference times. Training and inference times represent the wall-clock time per graph, measured on a single H100 GPU equipped with 8 AMD EPYC 9554 CPUs. For the POKEC dataset, due to its large graph size, inference times were computed entirely on CPUs. Note that training per graph can sometimes appear faster because we typically use a larger sampling rate during inference to achieve better performance. Specifically, for Benchmarking GNNs, LRGB, and OGBG datasets, the sampling rate is fixed at 1.0 during inference while it is usually smaller than 1.0 during training as given in Table 9, 10, 11. For node classification datasets, the inference sampling rate is 5 or 10 times higher than the rate used during training, as shown in Table 12.

