# OpenReview forum: "Learning Long Range Dependencies on Graphs via Random Walks"
_ICLR.cc/2025/Conference — ICLR 2025 Poster_

### Official Review · Reviewer_ehiS · 2024-10-30

**Soundness:** 3
**Presentation:** 3
**Contribution:** 3
**Rating:** 8
**Confidence:** 4

**Summary:**

The paper proposes the NeuralWalker architecture. This GNN processes sequential graph features obtained from random walks with sequence models such as SSMs. This process is interleaved with message passing modules. This construction aims to address the main weaknesses of prior random walk-based methods by encouraging longer-range information flow and utilizing breadth-first structural information.

The paper provides a theoretical analysis showing formally that NeuralWalker is more expressive than 1-WL and CNN-based random walk models like CRaWl. An experimental analysis further demonstrates strong overall performance on a range of standard benchmarks. The option of pertaining on the task of predicting the random walk features is explored. The experiments further show that random-walk-based methods like NeuralWalker can scale to large graphs with over 1M vertices.

**Strengths:**

* NeuralWalker successfully addresses several weaknesses in prior random-walk-based approaches by using SSMs in a novel context.
* The provided experiments and ablations are rather comprehensive.
* Pretraining a sequence model to predict the structural encodings of random walks is a novel idea, and the results presented here seem promising.

**Weaknesses:**

A weakness of this paper is a lack of discussion of "old" RNN architectures like LSTMs. While these have become less popular, it would be helpful to highlight why more modern long-sequence models like SSMs are preferable in the context of NeuralWalker. In fact, when compared to both Transformers and SSMs classical RNNs have been shown to have superior expressivity in terms of circuit complexity when the input sequence length is unbounded [1]. In a setting where the walk length is unbounded, an RNN-based NeuralWalker may therefore be more expressive than an SSM-based model. Exploring such a scenario in-depth is out of scope for the work presented here, but I do think a brief discussion of RNNs as an alternative sequence model choice is justified.

[1] Merrill, William, Jackson Petty, and Ashish Sabharwal. "The Illusion of State in State-Space Models." Forty-first International Conference on Machine Learning.

**Questions:**

* The result that SSMs outperform Transformers in the context of NeuralWalker is very interesting. Are there some theoretical insights as to why self-attention is not well-aligned with processing walk embeddings?

* I would also appreciate a comment on the point raised in "Weaknesses".

---

> ### Author Response · Authors · 2024-11-19
> **Author response to Reviewer ehiS**
>
> Thank you for your encouraging words and helpful feedback. We have tried to address your points below and in the revised manuscript. If you have any further questions, please let us know.
>
> > A weakness of this paper is a lack of discussion of "old" RNN architectures like LSTMs.
>
> We appreciate you bringing attention to the important role of classical RNN architectures like LSTMs. We acknowledge that our focus on recent advancements (SSMs, Transformers), which offer more efficient implementations, led us to overlook these classical RNN models. This is particularly relevant given the paper you cited \[Merrill et al.\] regarding RNNs' superior expressivity for unbounded sequences. Based on your suggestion, we have included a discussion in Section B.3 in the revised manuscript.
>
> > The result that SSMs outperform Transformers in the context of NeuralWalker is very interesting. Are there some theoretical insights as to why self-attention is not well-aligned with processing walk embeddings?
>
> The superior performance of SSMs over Transformers in our context was indeed an interesting empirical finding. While a complete theoretical analysis is beyond our current scope, we can offer several insights:
> - We maintained a strict parameter budget of ~500K following previous work. However, transformers typically require more parameters than other sequence models to achieve optimal performance.
> - While Transformers are universal approximators of sequence-to-sequence functions (Yun et al., 2020), this theoretical capability may require larger model capacities than our parameter budget allowed.

---

> > ### Comment · Reviewer_ehiS · 2024-12-02
> >
> > I appreciate the authors response to my questions and the improvements made to the manuscript. I will maintain my score of 8 as I think this is a strong paper worthy of publication.

---

### Official Review · Reviewer_CxEX · 2024-11-03

**Soundness:** 2
**Presentation:** 3
**Contribution:** 2
**Rating:** 6
**Confidence:** 4

**Summary:**

The paper presents NeuralWalker, which combines random walks and message-passing mechanisms to capture both local and long-range dependencies in graphs. The model processes graphs by treating random walks as sequences, thereby leveraging sequence models to handle long-range interactions.

**Strengths:**

1.	Existing graph transformers works typically incorporate random walks as positional encodings; directly integrating them into token design is an interesting approach.

2.	The theoretical analysis provided is thorough and comprehensive.

3.	Extensive experiments are conducted, with detailed results presented both in the main text and the appendix.

**Weaknesses:**

1.	The primary concern is the unclear motivation. Why do the authors choose to integrate random walks into token design, and how could this approach benefit over graph transformers, which can already use positional encodings to capture structural information? A preliminary study exploring this choice would be helpful.

2.	Building on this, the purpose of incorporating multiple techniques, such as the walk aggregator, also lacks clarity. For instance, why use average pooling in the walk aggregator? Do different random walks carry the same importance?

3.	Another concern is the lack of technical innovation. It’s difficult to discern a clear distinction between NeuralWalker and graph transformers, as NeuralWalker appears to follow general design principles of graph transformers, treating the graph as a sequence and representing tokens as the combination of nodes, edges, and positional encodings, albeit based on random walks.

4.	Could the authors clarify why SSE performs better than transformers? Since SSE and Mamba are designed based on RNNs, have bidirectional RNNs been tried as a sequence model?

5.	I also question the motivation for including global message passing although the ablation study shows it could help. When the length of random walks is long, they should already capture global information inherently. Besides, the authors directly borrow existing graph transformer methods for this, could you provide an ablation study on the choice of graph transformer in this part?

6.	Following this, an empirical efficiency comparison—specifically running time and memory usage—against current graph transformer baselines like Polynormer would be beneficial.

7.	Another major issue is the absence of several recent baselines. In traditional GNNs, several works, like LazyGNN[1], aim to capture long-range dependencies. There are also state-of-the-art graph transformers, such as VCR-Graphormer[2] and GOAT[3], and crucially, since the authors use SSE as a sequence model, Graph Mamba[4] should be included as a baseline.

8.	The datasets are not large enough, especially for node classification tasks. Commonly used datasets like the medium-sized ogbn-products or the large-sized ogbn-papers100m are not included. The current datasets are not common and cannot be considered as large-scale.

[1] LazyGNN: Large-Scale Graph Neural Networks via Lazy Propagation

[2] VCR-Graphormer: A Mini-batch Graph Transformer via Virtual Connections

[3] GOAT: A Global Transformer on Large-scale Graphs

[4] Graph-Mamba: Towards Long-Range Graph Sequence Modeling with Selective State Spaces

**Questions:**

1.	Is the motivation for using the random walk sampler driven by memory constraints? Additionally, in Figure 3, why does the ZINC dataset display a different trend compared to the other two datasets?

---

> ### Author Response · Authors · 2024-11-19
> **Author response to reviewer CxEX**
>
> Thank you for your time and detailed feedback. We have addressed your concerns below and in the revised manuscript. If you have further questions, please let us know.
>
> W1. Graph transformers compress random walks into fixed-size vectors, which either limits their structural expressivity or makes them difficult to analyze theoretically (Zhu et al. 2023). Our approach of treating random walks as sequences is theoretically motivated to avoid this limitation. Please see our (i) Technical contribution section for a detailed discussion of this motivation.
>
> W2. Our choice of average pooling in the walk aggregator is deliberate for two key reasons: 1) It aligns with our theoretical analysis, maintaining the connection between practice and theory. 2) It enables flexible deployment: the number of random walk samples can be adjusted between training and inference to offer a trade-off between performance and computation. In our experiments, we always use a sampling rate of 1.0 at inference but it was selected based on the dataset size during training.
>
> W3. Please refer to our general response section on (i) Technical contributions for a detailed discussion.
>
> W4. While Mamba empirically outperformed transformers as the sequence layer in our experiments, we acknowledge that a comprehensive analysis of this phenomenon is beyond our scope. However, we can offer an insight: the performance difference may stem from parameter constraints (500K budget) as we strictly followed the configurations of previous works. Transformers typically require more parameters than other sequence models such as CNNs or Mamba to achieve optimal performance.
>
> Regarding using bidirectional RNNs as the sequence layer, we would like to point out that SSMs are also RNN models. While exploring classical RNNs in depth could be an interesting further research direction, we want to emphasize that we focused on the most recent advances in RNNs (such as Mamba and S4), which offer more efficient implementations. As suggested by Reviewer eHiS, RNNs could indeed be useful for unbound sequences, we have thus added a discussion in Section B.3 in the revised manuscript.
>
> W5. While long random walks theoretically capture long-range information (as proven in our theory for subgraph isomorphism test), we found empirical benefits from including global message passing for certain datasets since the walk length cannot be arbitrarily long in practice (the expressive power of NeuralWalker with length $\ell$ is also given in our Thm. 4.4). As a consequence, we use a global message passing to compromise this in practice. The global message passing strategies considered in our work are the virtual node module and the vanilla transformer layer (**rather than a full graph transformer architecture**) and an ablation study is provided in Table 6a. Please refer to the paragraph "Effect of local and global message passing" in Section 5.3 for more details.
>
> W6. Our space complexity matches the time complexity proven in Theorem 4.5, with empirical validation shown in Figure 3. A key advantage of our method over graph transformers is explicit control over memory usage through sampling rates and walk lengths. For example, while Polynormer struggled with datasets of very large graphs like Pokec (they deactivated the attention module in their [code](https://github.com/cornell-zhang/Polynormer/blob/fc8c276c9c5dfbd616d83f65338a3392188a5e08/large_graph_exp/lg_model.py#L81)), our method successfully scaled to this size.
>
> W7. Please see the (ii) Additional baselines section in our general response.
>
> W8. Our work presents a general graph representation learning framework, not specifically a node classification method. Our evaluation spans 19 diverse benchmarks, from synthetic to large-scale real-world datasets, covering graph, node, and edge prediction tasks. The inclusion of node classification datasets specifically demonstrates our method's scalability advantage over current transformer-based approaches, which often struggle with such scales. Future work could certainly explore additional larger scale datasets, but we believe our current evaluation provides strong empirical support for our method's effectiveness and generalizability across diverse graph learning scenarios. Our theoretical analysis on the complexity also provides sufficient promise of our model's scalability to larger datasets.
>
> > Q1. Is the motivation for using the random walk sampler driven by memory constraints?
>
> The role of the random walk sampler is to independently sample a subset of random walks on each graph. We kindly point the reviewer to Section 3.2.
>
> > Q2. in Figure 3, why does the ZINC dataset display a different trend compared to the other two datasets?
>
> The apparent difference in ZINC's trend is due to its use of MAE as the evaluation metric for regression. The underlying performance pattern aligns with other datasets when accounting for this metric difference.

---

> > ### Comment · Reviewer_CxEX · 2024-11-24
> >
> > Thank you for the author's detailed response. However, some of my concerns remain:
> >
> > 1. The novelty still seems limited. Random walks are widely used in graph transformers to capture global information, and message passing is typically paired with them to capture local information since local neighbors are important for tasks such as node classification. Therefore, I do not see significant technical innovations in this work.
> >
> > 2. From Table 6(a), it appears that using a transformer for global message passing only benefits 1 out of 3 datasets. Given the additional computational cost introduced by the transformer, the necessity of this component is unclear.
> >
> > 3. I did not see any empirical memory cost in Figure 3. Additionally, a theoretical space complexity comparison with other baselines would be beneficial to demonstrate the advantages of this work.
> >
> > 4. I do not quite agree with the response regarding point w8. I believe the author intends to use node classification results to further showcase the advantages of the proposed methods. Therefore, I suggest that the author at least use some commonly used datasets to make the comparison clearer to all readers.
> >
> > 5. I still expect to see comparisons with other state-of-the-art GNN and graph transformer works. I believe these models are not specific to node classification, and downstream tasks should not limit the comparison with these models.
> >
> > Given the current response from the author, I would like to maintain my rating until these concerns are addressed.

---

> > > ### Author Response · Authors · 2024-11-25
> > > **Author response to Reviewer CxEX (1/2)**
> > >
> > > > 1. The novelty still seems limited. Random walks are widely used in graph transformers to capture global information, and message passing is typically paired with them to capture local information since local neighbors are important for tasks such as node classification. Therefore, I do not see significant technical innovations in this work.
> > >
> > > We agree that using information from random walks, in the form of structural or positional encodings (fixed-sized vectors), is a well-established concept. However, we would like to stress that the key innovation of our approach lies in leveraging random walks as sequences with the proposed positional encoding. This encoding preserves full induced subgraph information and ensures the subgraph isomorphism test power. Combined with message passing, our model surpasses the expressivity of both the 1-WL test and the ($\lfloor \ell/2 \rfloor + 1$)-subgraph isomorphism test. **To the best of our knowledge, this level of expressivity has not been demonstrated by existing graph transformers**. Furthermore, **our expressivity results do not rely on the use of the global message passing mechanism**, making our work fundamentally different from graph transformers. We are not aware of any prior work explicitly using random walks as sequences within a graph transformer. If there are any relevant works that we might have overlooked, we would be grateful for any references you could provide.
> > >
> > > We believe these theoretical insights and the novel integration of random walks and message passing highlight the technical contributions of our work. Please let us know if there are additional concerns or specific aspects you would like us to elaborate on further.
> > >
> > > > 2. From Table 6(a), it appears that using a transformer for global message passing only benefits 1 out of 3 datasets. Given the additional computational cost introduced by the transformer, the necessity of this component is unclear.
> > >
> > > We would like to clarify that we do not claim the transformer component is essential. As explicitly stated in Section 3.3.4, "Following the local message passing layer, we optionally apply a global message passing mechanism, allowing for global information exchange." In fact, in the majority of datasets we evaluated (16 out of 19, as shown in Tables 9, 10, 11, 12), we found that either avoiding global message passing entirely or using virtual nodes as a lightweight alternative was sufficient, both of which scale at most linearly with the number of nodes. Even when employing a transformer-based global message passing mechanism, the memory requirements scale linearly with the sequence length when using FlashAttention (see [this](https://arxiv.org/pdf/2205.14135)).
> > >
> > > Additionally, our theoretical analysis does not rely on any form of global message passing to establish the model's expressivity. The optional use of the transformer component allows flexibility for cases where global information exchange might provide additional benefits, as seen in certain datasets. However, the inclusion of global attention is not a requirement for a strong performance across most tasks. The use of global attention is completely flexible, depending on the size of the dataset.
> > >
> > > Moreover, we believe our findings on the effects of global message passing offer valuable insights into designing more effective global message passing mechanisms that can strictly outperform existing approaches.
> > >
> > > > 3. I did not see any empirical memory cost in Figure 3. Additionally, a theoretical space complexity comparison with other baselines would be beneficial to demonstrate the advantages of this work.
> > >
> > > Following your suggestion, we have measured and reported the GPU memory usage of NeuralWalker on the ZINC, CIFAR10, and PascalVOC-SP datasets. Detailed results and analyses can be found in Section D.4 and Figure 5. In summary, our findings show that NeuralWalker scales strictly linearly with both the sampling rate and length, irrespective of the global message-passing mechanism employed. This scalability underscores the model’s potential for efficient application to very large datasets. A discussion on the theoretical space complexity of NeuralWalker is provided in Section C.3. Due to the **significant methodological differences** in baselines and NeuralWalker, a direct comparison is not straightforward and falls outside the scope of this work.

---

> > > > ### Author Response · Authors · 2024-11-25
> > > > **Author response to Reviewer CxEX (2/2)**
> > > >
> > > > > 4. I do not quite agree with the response regarding point w8. I believe the author intends to use node classification results to further showcase the advantages of the proposed methods. Therefore, I suggest that the author at least use some commonly used datasets to make the comparison clearer to all readers.
> > > >
> > > > We would like to emphasize that we consider the node classification task merely as an additional evaluation that highlights the universality of our method. This specific task was not the focus of this work, which we tried to clarify by the statement in Section 5.1: "Node classification on large graphs: We **further** explored NeuralWalker’s ability to handle large graphs in node classification tasks". Thus, the goal of including experiments on node classification was to provide a preliminary evaluation rather than an exhaustive analysis. To that end, we followed the experimental settings and choices provided by Polynormer without extensive hyperparameter tuning.
> > > >
> > > > Nevertheless, in response to your suggestion, we have included preliminary results on the ogbn-products dataset, a well-known homophilic dataset comprising about 2M nodes and 61M edges. NeuralWalker was evaluated using the same experimental setup as for the Pokec dataset to ensure consistency. With minimal hyperparameter tuning, our model slightly outperforms Polynormer in terms of average accuracy over 10 runs.
> > > >
> > > > | Model         | GraphGPS | Exphormer | LazyGNN    | NAGPHORMER   | Polynormer   | NeuralWalker |
> > > > | ------------- | -------- | --------- | --- | ------------ | ------------ | ------------ |
> > > > | OGBN-Products | OOM      | OOM       | 82.3    | 73.55 ± 0.21 | 83.82 ± 0.11 | 83.85 ± 0.13 |
> > > >
> > > > > 5. I still expect to see comparisons with other state-of-the-art GNN and graph transformer works. I believe these models are not specific to node classification, and downstream tasks should not limit the comparison with these models.
> > > >
> > > > Following your suggestions, we have added Graph-Mamba in Table 2. We have also compared NeuralWalker to GOAT below where its results are available from Polynormer (Deng et al., 2024). We have also included LazyGNN in the above ogbn-products benchmark. However, VCR-Graphormer was not evaluated on any of the datasets used in our work. Due to its significant methodological differences from NeuralWalker and its focus on node classification, we believe generalizing it to general downstream tasks is beyond the scope of our work. Thus, we decided not to include it in our comparisons.
> > > >
> > > > | Model        | ROMAN-EMPIRE | AMAZON-RATINGS | MINESWEEPER  | TOLOKERS     | QUESTIONS    | POKEC        |
> > > > | ------------ | ------------ | -------------- | ------------ | ------------ | ------------ | ------------ |
> > > > | GOAT         | 71.59 ± 1.25 | 44.61 ± 0.50   | 81.09 ± 1.02 | 83.11 ± 1.04 | 75.76 ± 1.66 |  66.37 ± 0.94 |
> > > > | NeuralWalker | 92.92 ± 0.36 | 54.58 ± 0.36   | 97.82 ± 0.40 | 85.56 ± 0.74 | 78.52 ± 1.13 | 86.46 ± 0.09 |

---

> > > > > ### Comment · Reviewer_CxEX · 2024-11-28
> > > > >
> > > > > I have checked your response to me and area chair, so I decided to raise my score.

---

> > > > > > ### Author Response · Authors · 2024-11-28
> > > > > > **Thank you**
> > > > > >
> > > > > > Thank you again for your time and effort in evaluating our work! We are happy to continue the discussion if you have any remaining questions or doubts.

---

### Official Review · Reviewer_J8Bn · 2024-11-03

**Soundness:** 3
**Presentation:** 4
**Contribution:** 3
**Rating:** 8
**Confidence:** 4

**Summary:**

In order to learn long range dependencies, this paper presents NeuralWalker architecture that samples random walks from the graph, treats them as sequences, encode these sequences using a sequence layer and finally integrates this module with local and global message passing to better learn graph structure. It provides theoretical results to show the expressive power of NeuralWalker.

**Strengths:**

- The paper is overall well-written.
- Incorporating random-walks in graph learning is an important direction mainly due to its efficiency and ability to learn the long-range dependencies.
- Theoretical results have provided detailed discussions about the expressive power of NeuralWalker and motivates its design.
- The training details for reproducibility are reported, which can help future studies to better understand the weaknesses/strengths of NeuralWalker.

**Weaknesses:**

- My main concern is that there are several claims in the paper that have remained unsupported/unclear:
  - The authors claim that ``our approach achieves significant performance improvements on 19 graph and node benchmark datasets``. Based on the reported results NeuralWalker underperforms baselines and several missed state-of-the-art methods. Even looking at the current baselines NeuralWalker does not provide performance improvements in all the 19 datasets!
  -  The authors claim that ``CNNs present a compelling alternative for large datasets due to their faster computation (typically 2-3x faster than Mamba on A100)``. Where are the results for this? How is this trade-off between efficiency and performance?
  - The authors several times mentioned efficiency as the motivation of their method. I couldn’t find any comparison with baselines with respect to memory usage and speed. While training time is reported in Tables 9-12, the training time for baselines and also the memory usage of NeuralWalker and baselines are unclear.
  - Without local/global encoding the performance of NeuralWalker significantly drops. How the results can support the importance of using random walks for long-range dependencies?
  - Since the authors have claimed performance improvement over existing methods, it is important to choose the right baselines. For example, although the results of POLYNORMER (Deng et al., ICLR 2024) are reported in Table 5, their results are missing in other experiments. Also, the results for several recent efficient methods are missing: GSSC (Huang et al., 2024), GOAT (Kong et al., ICML 2023), and GRED (Ding et al., ICML 2024). It would be great if the authors could use consistent baselines across different experiments so the performance gain becomes clear. Also, I believe a comparison with random walk-based GNNs like GraphSAGE is required.
- There are several missing related articles. I suggest discussing these methods to further clarify the novelty of NeuralWalker. For example, there are several recent methods that also have used sequence layers for graphs GSSC (Huang et al., 2024), and GRED (Ding et al., ICML 2024).
- This paper has overlooked a version of graph-mamba (Behrouz et al., KDD 2024), which also uses random walks to learn long-range dependencies. I found NeuralWalker very similar to this method when the number of sampled walks per node is 1. It would be great if the authors clarify and discuss the differences.
- Based on the results in Table 6, without local and global message passing the performance of the proposed method drops significantly. This raises concerns about the significance of the proposed method as it seems the main reason for the competitive performance of NeuralWalker is the use of local/global message-passing modules.

Please also see the questions.

**Questions:**

- Why Polynormer’s results are missing in LRGB and GNN benchmark experiments?
- What are the main differences between NeuralWalker with the tokenization module in graph-mamba (Behrouz et al., 2024), where the number of random walk samples ($M$) in their framework is 1?
- Although random walks can provide a natural sequence to learn long-range dependencies, they also can be sensitive to noisy connections. How does NeuralWalker perform in these cases?
- In the theoretical results, don’t you need to assume that your sequence model is a universal approximator? Without this assumption how one can compare the expressive power of NeuralWalker after encoding walks with the 1-WL test?

---

> ### Author Response · Authors · 2024-11-19
> **Author response to reviewer J8Bn (1/2)**
>
> Thank you for your time and detailed feedback. We have addressed your concerns below and in the revised manuscript. If you have further questions, please let us know.
>
> > The authors claim that `our approach achieves significant performance improvements on 19 graph and node benchmark datasets`. Based on the reported results NeuralWalker underperforms baselines and several missed state-of-the-art methods.
>
> We apologize for the unclear language regarding performance improvements. We have revised our statement to "our approach achieves competitive performance on 19 graph and node benchmark datasets". Regarding your concern on missing methods, please refer to our answer (ii) Additional baselines in our general response above.
>
> > The authors claim that CNNs present a compelling alternative for large datasets due to their faster computation (typically 2-3x faster than Mamba on A100). Where are the results for this? How is this trade-off between efficiency and performance?
>
> The performance comparison between CNN and Mamba is presented in Table 6b. While we observed CNNs to be 2-3x faster than Mamba on an A100, with training times reported in Tables 9, 10, 11, and 12, we omitted detailed time comparison results due to space constraints. This comparison primarily serves to justify our use of CNN for large-scale datasets in OGB, rather than being a central claim of our work. Based on your suggestion, we have included the time comparison results in Table 14 in the Appendix.
>
> > The authors several times mentioned efficiency as the motivation of their method.
>
> We want to clarify that efficiency is not our primary motivation. Our method is fundamentally motivated by the complementary nature of random walks (depth-first) and message passing (breadth-first) in graph exploration (Please see (i) Technical contribution section in our general response). On the other hand, our discussion of efficiency focuses on the flexibility to balance model complexity and expressivity through walk sampling parameters, as presented in Section 5.3.
>
> > Without local/global encoding the performance of NeuralWalker significantly drops. How the results can support the importance of using random walks for long-range dependencies?
>
> > Based on the results in Table 6, without local and global message passing the performance of the proposed method drops significantly.
>
> The significant performance impact of local message passing actually supports our core thesis: the synergy between random walks and message passing is crucial for achieving superior expressivity. This is both theoretically proven and empirically demonstrated in our work (please also see (i) Technical contributions section in our general response regarding this aspect).
>
> > although the results of POLYNORMER (Deng et al., ICLR 2024) are reported in Table 5, their results are missing in other experiments.
>
> > Why Polynormer’s results are missing in LRGB and GNN benchmark experiments?
>
> We included Polynormer results where directly comparable, noting that its primary focus is on extending the scalability of graph transformers for node classification on large graphs. Polynormer aims to balance expressivity and scalability for GTs rather than improve expressivity. Extending it to graph-level or edge-level tasks falls outside our paper's scope.
>
> > Adding other baselines such as GSSC, GOAT, and GRED.
>
> Please refer to our discussion of (ii) Additional baselines in our general response, which explains our focus on methods directly comparable to our key innovations.
>
> > I believe a comparison with random walk-based GNNs like GraphSAGE is required.
>
> GraphSAGE uses random walks to design their loss function rather than explicitly encoding them into graph representations. Based on your suggestion, we have included a comparison with it in Table 5.
>
> > The paper has overlooked graph-mamba (Behrouz et al., KDD 2024). What are the differences between Graph-Mamba and your work?
>
> > What are the main differences between NeuralWalker with the tokenization module in graph-mamba (Behrouz et al., 2024), where the number of random walk samples (M) in their framework is 1?
>
> Please note that we did not overlook this version of graph-mamba, and we discussed the key difference in the "sequence modeling" paragraph in Section 2.
> Following your comment, we provide a more detailed discussion of the key differences between our approach and Graph-Mamba in the (ii) Additional baselines section of our general response.

---

> > ### Author Response · Authors · 2024-11-19
> > **Author response to Reviewer J8Bn (2/2)**
> >
> > > Although random walks can provide a natural sequence to learn long-range dependencies, they also can be sensitive to noisy connections. How does NeuralWalker perform in these cases?
> >
> > Thank you for raising this interesting question. While a comprehensive analysis of noise robustness is beyond our current scope, our method's use of multiple random walks provides inherent resilience: the aggregate information from multiple walks helps dilute the impact of noisy connections when sufficiently clean connections exist. The node proximity information is captured through multiple random walks, reducing sensitivity to individual noisy paths.
> >
> > > In the theoretical results, don’t you need to assume that your sequence model is a universal approximator? Without this assumption how one can compare the expressive power of NeuralWalker after encoding walks with the 1-WL test?
> >
> > You are correct that we assume the sequence model is a universal approximator ($\mathcal{F}$ is universal). To clarify the expressivity comparison: the universality of the sequence model ensures that NeuralWalker's expressivity exceeds the subgraph isomorphism test, while the message passing component guarantees the 1-WL expressivity. These complementary properties contribute to our method's overall expressivity.

---

> > > ### Comment · Reviewer_J8Bn · 2024-11-24
> > >
> > > I thank the authors for their response. I believe the paper has interesting and valuable contributions, and so I have updated my scores. However, it is notable that I still think there are some parts that need improvements. So I summarized them as follows, I hope it helps the authors to further improve the paper:
> > >
> > > 1. It would be beneficial if the authors could provide the results on the memory usage of NeuralWalker. It can help readers to understand the advantages and disadvantages of NeuralWalker.
> > > 2. I still believe it is an important drawback for NeuralWalker that without local/global encoding its performance drops significantly. It would be beneficial for the paper to be honest about this drawback and discuss it in detail.
> > > 3. (Minor) In the baselines section, GRED is written as GREG.
> > > 4. (minor) As I mentioned in my initial review, there is a version of graph mamba that does not need node ordering and is based on random walks. The current discussion (line 135) is somehow misleading.

---

> > > > ### Author Response · Authors · 2024-11-25
> > > > **Author response to Reviewer J8Bn**
> > > >
> > > > Thank you for considering our rebuttal and manuscript revisions, as well as your encouraging words! We provide below responses to your additional comments:
> > > >
> > > > > 1. It would be beneficial if the authors could provide the results on the memory usage of NeuralWalker. It can help readers to understand the advantages and disadvantages of NeuralWalker.
> > > >
> > > > Following your suggestion, we have measured and reported the GPU memory usage of NeuralWalker on the ZINC, CIFAR10, and PascalVOC-SP datasets. Detailed results and analyses can be found in Section D.4 and Figure 5. In summary, our findings show that NeuralWalker scales strictly linearly with both the sampling rate and length, irrespective of the global message-passing mechanism employed.
> > > >
> > > > > 2. I still believe it is an important drawback for NeuralWalker that without local/global encoding its performance drops significantly. It would be beneficial for the paper to be honest about this drawback and discuss it in detail.
> > > >
> > > > We would like to clarify that local message passing does not significantly increase the space and time complexity of our method. As discussed in our theorem 4.5 and Section C.3, the time and space complexity of computing local message passing is dominated by the complexity of processing random walks with Mamba layers, provided the graph is not extremely dense ($\beta<\gamma \ell$).
> > > >
> > > > Additionally, the use of global message passing is optional, as its impact on performance varies across datasets--a phenomenon also observed by Rosenbluth et al. (2024). Notably, in the majority of datasets we evaluated (16 out of 19), avoiding global message passing or simply employing virtual nodes as the global message passing mechanism was sufficient to achieve a strong performance while maintaining linear scalability with the number of nodes. Even when employing a transformer-based global message passing mechanism, the memory requirements scale linearly with the sequence length when using FlashAttention (see [this](https://arxiv.org/pdf/2205.14135)).
> > > >
> > > > We believe our findings on the effects of global message passing should not be seen as a drawback of NeuralWalker. Instead, they offer valuable insights into designing more effective global message passing modules that can strictly outperform existing approaches.
> > > >
> > > > Please let us know if there are specific areas you would like us to elaborate on further, or if you have any concrete suggestions you would like us to incorporate.
> > > >
> > > > > 3. (Minor) In the baselines section, GRED is written as GREG.
> > > >
> > > > Thank you for spotting this typo. We have fixed it accordingly in the revised manuscript.
> > > >
> > > > > 4. (minor) As I mentioned in my initial review, there is a version of graph mamba that does not need node ordering and is based on random walks. The current discussion (line 135) is somehow misleading.
> > > >
> > > > Thank you for the clarification, and we have updated the manuscript accordingly to clarify this point (Section 2, "Sequence modeling").

---

> > > > > ### Comment · Reviewer_J8Bn · 2024-11-26
> > > > >
> > > > > I appreciate the authors' effort in the rebuttal and thank them for their detailed response and for addressing all my concerns. I do not have any additional questions/concerns. As I mentioned before, I believe this paper has valuable contributions and interesting results, and the current version is in good shape. I have updated my scores to reflect this.

---

> > > > > > ### Author Response · Authors · 2024-11-26
> > > > > > **Thank you**
> > > > > >
> > > > > > Thank you for your timely reply and support! We are happy to continue the discussion if you have any other questions or doubts.

---

### Official Review · Reviewer_p6gD · 2024-11-04

**Soundness:** 3
**Presentation:** 3
**Contribution:** 2
**Rating:** 6
**Confidence:** 4

**Summary:**

This paper addresses the challenge of capturing long-range dependencies in GNNs by combining random walks with local message passing. The authors utilize State Space Models (SSMs) to model random walk sequence. Theoretical analysis highlights the expressiveness of the proposed approach, and experimental results show that it outperforms the selected baselines.

**Strengths:**

1. The issue of long-range dependencies in GNNs is an important and valuable area of exploration.
2. The proposed method is versatile and can be integrated with various models.
3. The authors offer theoretical insights, including the expressiveness of the proposed method.

**Weaknesses:**

1. The novelty of this method is limited, as random walks, combining local and global information, and SSMs are established techniques. For example, Graph-Mamba [1] also uses random walks and Mamba on GNNs, and graph transformers typically combine global and local data.

2. The datasets selected for node classification are mainly heterophilic. Including commonly used datasets like OGB would provide a more comprehensive evaluation.

3. The source of the performance gains is unclear—whether they stem from pretraining, positional encodings, or SSMs. An ablation study would help clarify each component's impact.

[1] Behrouz, Ali, and Farnoosh Hashemi. "Graph mamba: Towards learning on graphs with state space models." Proceedings of the 30th ACM SIGKDD Conference on Knowledge Discovery and Data Mining. 2024.

**Questions:**

Please refer to the weaknesses.

---

> ### Author Response · Authors · 2024-11-19
> **Author response to Reviewer p6gD**
>
> Thank you for your time and detailed feedback. We have addressed your concerns below and in the revised manuscript. If you have further questions, please let us know.
>
> > Novelty is limited
>
> While we agree that methods such as random walks, message passing, and SSMs are established techniques, we believe our approach introduces significant novelty by successfully combining them into a method that achieves state-of-the-art performance with several theoretically relevant properties. Please refer to the (i) Technical contributions above for a discussion on our contribution and its novelty. Furthermore, we discuss a detailed comparison between Graph-Mamba and our method in (ii) Additional baselines above.
>
> > The datasets for node classification are mainly heterophilic.
>
> We appreciate the reviewer's observation about the heterophilic datasets. However, our work presents a general graph representation learning framework, not specifically a node classification method. Our evaluation spans 19 diverse benchmarks, from synthetic to large-scale real-world datasets, covering graph, node, and edge prediction tasks. The inclusion of node classification datasets specifically aims to demonstrate our method's scalability advantage over current transformer-based approaches, which often struggle with such scales. Future work could certainly explore additional datasets, but we believe our current evaluation provides strong empirical support for our method's effectiveness and generalizability across diverse graph learning scenarios.
>
> > The source of the performance gains is unclear
>
> The sources of our performance improvements are systematically studied and documented in Sections 5.2 and 5.3. Section 5.2 provides empirical evidence of pretraining benefits on the ZINC dataset, while Section 5.3 offers detailed ablation studies on message passing and sequence layer architectures. Furthermore, our random walk positional encodings are not merely an implementation detail but a theoretical cornerstone, enabling our proven $(\lfloor\ell/2 \rfloor + 1)$-subgraph isomorphism expressiveness (Section 4). We have added a clarification sentence in Section 3.2.

---

> > ### Comment · Reviewer_p6gD · 2024-11-23
> >
> > Thanks for the author's reply. Although the author claims the proposed method is evaluated on node, edge, and graph-level tasks,  they should select the most popular datasets and settings.  I will keep my rating.

---

> > > ### Author Response · Authors · 2024-11-25
> > > **Author response to Reviewer p6gD**
> > >
> > > Thank you for considering our rebuttal and manuscript revisions.
> > >
> > > We would like to highlight that the datasets used in our study are widely recognized in the field of graph learning. For instance, the Benchmarking GNNs, LRGB, and OGBG datasets have been employed in prominent works such as GraphGPS (Rampasek et al., 2022), Exphormer (Shirzad et al., 2023), and GRIT (Ma et al. 2023), among others.
> > >
> > > In response to your suggestion, we have included preliminary results on the ogbn-products dataset, a well-known homophilic dataset comprising about 2M nodes and 61M edges. NeuralWalker was evaluated using the same experimental setup as for the Pokec dataset to ensure consistency. With minimal hyperparameter tuning, our model slightly outperforms Polynormer in terms of average accuracy over 10 runs.
> > >
> > >
> > > |  Model       | GraphGPS | Exphormer | NAGPHORMER   | Polynormer   | NeuralWalker |
> > > | ------------- | -------- | --------- | ------------ | ------------ | ------------ |
> > > | ogbn-products | OOM      | OOM       | 73.55 ± 0.21 | 83.82 ± 0.11 | 83.85 ± 0.13 |

---

### Author Response · Authors · 2024-11-19
**General response (1/3)**

We sincerely thank all the reviewers for their time and great efforts in providing detailed feedback, which we felt has strengthened our paper. We have incorporated several important changes into the revised PDF, which can be seen in red.

We appreciate the reviewers recognizing several key strengths of our work:

- (Reviewer p6gD, J8Bn) The problem studied is important and valuable.
- (Reviewer p6gD, CxEX, ehiS) The proposed method is versatile, interesting, and novel, successfully addressing several weaknesses in prior random-walk-based approaches.
- (Reviewer p6gD, J8Bn, CxEX, ehiS) Theoretical results are comprehensive, providing detailed discussions about the expressive power of NeuralWalker and motivating its design.
- (Reviewer CxEX, ehiS) Extensive experiments are conducted, with detailed results presented.
- (Reviewer J8Bn) The paper is overall well presented and documented, with training details reported for reproducibility.

---

> ### Author Response · Authors · 2024-11-19
> **General response (2/3)**
>
> While we provide individual responses to specific comments and concerns below, we address here several general points of feedback:
>
> ### (i) Technical contributions
>
> We would like to clarify our technical contributions:
>
> 1. **(Novel combination of complementary graph exploration paradigms for graph learning)** Our primary contribution lies in combining random walks and message passing paradigms. This combination is theoretically motivated by their complementary exploration nature: Random walks exhibit depth-first exploration and message passing follows breadth-first patterns. We prove that this synergy enables greater expressiveness than methods using either paradigm alone. More precisely, random walks of length $\ell$ help our model achieve $(\lfloor\ell/2 \rfloor + 1)$-subgraph isomorphism test power while message passing achieves 1-WL test power. Note that 1-WL test is not necessarily dominated by the $(\lfloor\ell/2 \rfloor + 1)$-subgraph isomorphism test as the size of WL-unfolding subtrees can be arbitrarily large. Please see Thm 4.4 and its discussion below for more details. The importance of combining the two paradigms is also validated in our experiments in Table 6. We believe this combination offers an important insight for future graph learning research.
> 2. **(Explicit sequence treatment of random walks)** Our approach fundamentally differs from existing graph transformers and GNNs in its explicit treatment of random walk as sequences, and processes them with state-of-the-art sequence models. While graph transformers compress random walks into fixed-size vectors, limiting their expressivity or making a theoretical analysis more difficult (Zhu et al. 2023), our explicit sequence handling offers two key advantages. First, it enables us to leverage advances in sequence modeling (in our experiments, we found Mamba worked the best). Second, it allows for an explicit analysis of model expressivity, as demonstrated in Thm. 4.4. These theoretical benefits are further validated by our empirical results, which show improved performances over graph transformers (e.g. GraphGPS which typically uses a random walk structural encoding) on several datasets. Moreover, the strong performance of our model does not necessarily rely on the global message passing module (which is nevertheless the main building block for GTs), making our model more scalable than most GTs (by scaling linearly with the number and length of random walks, which can be explicitly controlled in practice).

---

> > ### Author Response · Authors · 2024-11-19
> > **General response (3/3)**
> >
> > ### (ii) Additional baselines
> >
> > We want to clarify that our work presents a general graph representation learning framework rather than targeting a specific task like node classification. Our evaluation spans 19 diverse benchmarks, from synthetic to large-scale real-world datasets, covering graph, node, and edge prediction tasks. We included node classification datasets specifically to demonstrate two key aspects: our method's flexibility in integrating other models (demonstrated using Polynormer) and its scalability advantages over existing transformer-based approaches such as GraphGPS, which often struggle at such scales. To ensure fair comparisons, **we focused on baseline methods that, like NeuralWalker, address general graph representation learning—specifically those supporting graph, node, and edge prediction tasks**.
> >
> > We thank the reviewers for their suggestions regarding baselines. We now discuss their relevance to our work.
> >
> > - LazyGNN, GOAT, VCR-Graphormer, Polynormer: 1. These methods are primarily designed for node classification tasks and extending their models to other graph learning tasks fall outside our paper's scope. 2. Among these, we included Polynormer in our baselines, as it demonstrates superior performance over the other three methods (Deng et al., 2024).
> > - Other models (GRED and GSSC): 1. While both models use recurrent neural networks, they do not use any form of random walk representations in their models. 2. Given that GSSC is still an ongoing work, we focused on GRED, which we have now included in Tables 1 and 2 of our revised manuscript. The additional comparison does not change our main conclusions.
> > - Graph-Mamba (Behrouz et al. 2024) employs Mamba as a global message passing module rather than for processing random walk sequences. Moreover, it does not use random walks explicitly: it works with subgraphs induced by random walks and their descriptors (e.g. Laplacian, aggregated node features), introducing additional complexity. In contrast, our method's direct treatment of random walks as sequences, combined with message passing, achieves higher expressivity through a cleaner, more principled design. Despite these significant technical differences, we have included Graph-Mamba in Tables 1 and 2 of our revised manuscript, as suggested by multiple reviewers, by noting that our main conclusions remain unchanged.
> >
> > We are open to including additional baselines in the final version if the reviewers consider it essential, though we believe our current evaluation effectively demonstrates the distinct advantages of our approach.

---

> ### Comment · Area_Chair_HcpU · 2024-11-26
> **Some further comments**
>
> Appreciate the authors' effort! After reading this paper, I have one comment and one question.
>
> **Regarding Related Works**
> To the best of my knowledge, there are earlier works that utilize walks and positional encoding for graph representation learning [1][2][3]. I suggest the authors include these earlier references and properly position them in the paper. These works predate Tonshoff et al. (2023b) which was cited as the primary inspiration for this paper.
>
> **Scalability of the Work**
> If the time is allowed, could the authors add a figure to illustrate the relationship between the number of walks required for optimal performance for each dataset v.s. the number of nodes in the graphs? Similarly, it would be helpful to include a figure showing how the optimal walk lengths vary with the number of nodes. While I appreciate that Figure 3 provides insights into the sampling rate for selected graph datasets, it would be more informative to understand how these metrics scale with graph size.
>
> [1] Anonymous Walk Embeddings, Ivanov & Burnaev, ICML 2018
> [2] Inductive Representation Learning in Temporal Networks via Causal Anonymous Walks, Wang et al., ICLR 2021
> [3] Algorithm and System Co-design for Efficient Subgraph-based Graph Representation Learning, Yin et al., VLDB 2022

---

> > ### Author Response · Authors · 2024-11-26
> > **Author response to Area Chair HcpU**
> >
> > Thank you for your time in reading our paper and providing feedback! We have addressed your points below:
> >
> > > **Regarding Related Works**
> >
> > Thank you for pointing out these related articles. All three papers precede CRaWL [4] and similarly utilized random walks with identity encodings for graph representation learning.
> > The key difference from CRaWL [4] is that these approaches did not include adjacency encodings, which are essential for capturing the full information of induced subgraphs. Moreover, similar to the difference between our work and CRaWL, they do not incorporate message passing and study the synergy between them, which is a fundamental aspect of NeuralWalker.
> >
> > We have revised our paper accordingly to discuss these works in Section 2.
> >
> > [1] Anonymous Walk Embeddings, Ivanov & Burnaev, ICML 2018
> > [2] Inductive Representation Learning in Temporal Networks via Causal Anonymous Walks, Wang et al., ICLR 2021
> > [3] Algorithm and System Co-design for Efficient Subgraph-based Graph Representation Learning, Yin et al., VLDB 2022
> > [4] Walking Out of the Weisfeiler Leman Hierarchy: Graph Learning Beyond Message Passing, Toenshoff et al., TMLR 2023
> >
> > > **Scalability of the Work**
> >
> > Due to the high computational cost of tuning these hyperparameters and the large number of datasets evaluated in our work, optimizing the sampling rate and length individually for each dataset was not feasible. Therefore, in our experiments, we employed the following strategy: Where applicable, we adopted the hyperparameter values from Tonshoff et al. (2023b). For other datasets, we selected their values based on the size and scale of the datasets, rather than performing per-dataset optimization.
> >
> > For most node classification tasks (except for Pokec due to its larger graph size), we fixed the sampling rate and length to 0.01 and 1000 respectively without tuning, as these experiments were intended to demonstrate our model's scalability. We have included a table below summarizing the selected values for the number and length of random walks for each dataset, as well as the product of the sampling rate and length. However, we found that this table does not provide significant insights into the relationship between the optimal values for these parameters and the number of nodes, as the values were chosen heuristically rather than optimized.
> >
> > However, we would like to highlight that, as shown in Figure 3, the performance saturates quickly when relatively small values of sampling rate and walk length are used. Specifically, for optimal values of $\gamma$ (sampling rate) and $\ell$ (walk length), we observe that $\gamma \ell \leq 100$ is sufficient to achieve near-optimal performance across all considered datasets. This indicates that increasing their product beyond a certain point does not lead to significant performance gains.
> >
> > This observation suggests that the empirical space and time complexity for optimal models may scale linearly with the graph size. In practical scenarios, even if available RAM is insufficient to achieve optimal performance, one can still maximize prediction performance by selecting the largest value of $\gamma \ell$ that can fit into the available RAM. Our approach effectively balances resource constraints with prediction accuracy.
> >
> > We hope our response has addressed your question. And we are happy to incorporate this discussion into our revision if you find it useful.
> >
> >
> > | Dataset         | (Avg.) Number of nodes |  Number of RWs $n\gamma$ |   Length of RWs $\ell$ |   $\gamma\ell$ |
> > |:----------------|:------------:|:------------:|:---------:|:-------:|
> > | ZINC            |    23.2        |      23.2   |       50 |   50   |
> > | OGBG-MOLPCBA    |    26          |      13     |       25 |   12.5 |
> > | PCQM-Contact    |    30.1        |      15.05  |       75 |   37.5 |
> > | MNIST           |    70.6        |      35.3   |       50 |   25   |
> > | CLUSTER         |   117.2        |      58.6   |      200 |  100   |
> > | CIFAR10         |   117.6        |      58.8   |       50 |   25   |
> > | PATTERN         |   118.9        |      59.45  |      100 |   50   |
> > | OGBG-CODE2      |   125.2        |      62.6   |      100 |   50   |
> > | Peptides-Func   |   150.9        |      75.45  |      100 |   50   |
> > | Peptides-Struct |   150.9        |      75.45  |      100 |   50   |
> > | OGBG-PPA        |   243.4        |     121.7   |      200 |  100   |
> > | COCO-SP         |   476.9        |     119.225 |      100 |   25   |
> > | PASCALVOC-SP    |   479.4        |     239.7   |      100 |   50   |
> > | MINESWEEPER     | 10000          |     100     |     1000 |   10   |
> > | TOLOKERS        | 11758          |     117.58  |     1000 |   10   |
> > | ROMAN-EMPIRE    | 22662          |     226.62  |     1000 |   10   |
> > | AMAZON-RATINGS  | 24492          |     244.92  |     1000 |   10   |
> > | QUESTIONS       | 48921          |     489.21  |     1000 |   10   |
> > | POKEC           |     1,632,803 |    1632.8   |      500 |    0.5 |

---

> > > ### Comment · Area_Chair_HcpU · 2024-11-27
> > >
> > > Thank the authors for the prompt response! It's indeed crucial to position these works appropriately within the research domain of walk-based graph representation learning.
> > >
> > > I greatly appreciate the effort in providing the table detailing the number and length of random walks (RWs). I feel sorry for the previous confusion. I fully understand the challenges of determining the optimal sampling rate and length due to the hyperparameter tuning effort required. When I mentioned this, I assumed that the testing performance reported in the paper for each dataset reflects approximately optimal results. Therefore, the authors could simply report the numbers and lengths of RWs used to achieve the testing performance already presented in the paper—this should not require additional fine-tuning.
> > >
> > > Additionally, if the authors could provide the per-graph wall-clock times split into sampling these walks, preprocessing them, and running model inference, it would make the scaling results even clearer. Reporting the wall clock time for inference is enough. No need for the time for training, which should simplify this process.
> > >
> > > Alternatively, if the authors can report the testing performance and wall-clock times based on the RW parameters provided in the table in the above response, that would also be okay. However, this might be more challenging, as the model would need proper tuning to ensure good testing performance, which could be time-consuming.
> > >
> > > Ultimately, I hope the final version of the paper could include a table that demonstrates the actual scalability of the proposed model. Such a table is essential for providing the community with a comprehensive view of the pros and cons of walk-based methods.

---

> > > > ### Author Response · Authors · 2024-11-27
> > > > **Author response to Area Chair HcpU**
> > > >
> > > > Thank you for your thoughtful and detailed comment! Following your suggestion, we have added Section D.6 and Table 16 to our revised manuscript. These additions provide detailed statistics on the number and length of random walks selected for each dataset, as well as the corresponding training and inference times. The results demonstrate that random walk sampling is highly efficient, and typically faster than model training or inference. Additionally, our model exhibits reasonable scalability with increasing graph size.
> > > >
> > > > We interpreted your suggestion on including the preprocessing time of random walks as the time it takes to compute the random walk encodings, which we implemented within the random walk sampler and thus included in the sampling time. Please let us know if this does not properly address your suggestion.
> > > >
> > > > We appreciated your alternatice suggestion to furthermore include test performances in this table. However, in order to avoid duplicate information, we decided not to include them, by noting that test performance metrics are already comprehensively discussed in Table 15 and Section D.5.
> > > >
> > > > We believe this new table offers a more comprehensive perspective on the scalability of our  (and any walk-based) method, and that it has further strengthened the manuscript.
> > > >
> > > > Thank you again for your valuable feedback. We would be happy to address any additional questions or concerns you might have.

---

> > > > > ### Comment · Area_Chair_HcpU · 2024-11-27
> > > > >
> > > > > Thank you for the prompt reply! I just have a quick clarification: Are the wall clock times measured as an average per graph or per node? Interestingly, I noticed that training sometimes seems faster than inference.
> > > > >
> > > > > Aside from this clarification, I don’t have any other questions. Thanks again for addressing my questions!

---

> ### Author Response · Authors · 2024-11-27
> **Author response to Area Chair HcpU**
>
> The wall-clock times are measured as averages **per graph**. Training can sometimes appear faster because we typically use a **larger sampling rate during inference to achieve better performance** (which is another advantage of walk-based methods). Specifically, for Benchmarking GNNs, LRGB, and OGBG datasets, the sampling rate is fixed at 1.0 during inference while it is usually smaller than 1.0 during training. For node classification datasets, the inference sampling rate is 5 or 10 times higher than the rate used during training (where these values have already been provided in Table 12).
>
> Thank you again for your valuable feedback!

---

### Comment · Area_Chair_HcpU · 2024-11-23
**Reminder: Please Review Author Responses**

Dear Reviewers,

As the discussion period is coming to a close, please take a moment to review the authors’ responses if you haven’t done so already. Even if you decide not to update your evaluation, kindly confirm that you have reviewed the responses and that they do not change your assessment.

Thank you for your time and effort!

Best regards,
AC

---

### Meta-Review · Area_Chair_HcpU · 2024-12-12

**Metareview:**

The paper introduces NeuralWalker, a random-walk-based graph representation method. NeuralWalker models graphs as sequences derived from random walks, encoding them through sequence layers, and integrates these encodings with local and global message-passing mechanisms. The authors provide theoretical analyses showcasing the model's expressiveness, particularly its ability to surpass 1-WL expressivity. Experimental results on multiple benchmark datasets demonstrate competitive performance, suggesting that NeuralWalker is an efficient and scalable approach for graph representation learning.

**Strengths**

While random walks and SSMs are established techniques, the integration of random walks into the token design and subsequent use of SSMs is an interesting approach. In particular, using the set of random walks to represent graphs is still under-explored, compared to the commonly-used GNN-based approaches.

The authors also demonstrate the efficiency and scalability of this type of approach. Moreover, the empirical performance of this method looks strong.

Exploring the potential of pretraining random walk structural encodings is a promising direction.

**Weaknesses**

The methodology, while thoughtfully combined, is essentially a complex integration of existing approaches. The developed theory (expressive power is expected to be higher than 1-WL) is standard and does not indicate any out-of-expected results.

The reviewers also generally posted some concerns about the used datasets by arguing that they are not sufficiently diverse or large-scale. The authors have done extensive evaluations and include back-up evaluations during the discussion period.

Overall, I think the pros of this work overweight its cons by showing another successful pipeline that uses random-walk-based graph representation. So, I lean towards acceptance.

**Additional Comments On Reviewer Discussion:**

In the discussion, the authors effectively addressed most of the reviewers' concerns regarding insufficient experiments and missing baselines. As a result, several reviewers raised their evaluation scores.

Another concern, raised by multiple reviewers, is about the novelty of the proposed approach, as the framework primarily integrates existing techniques. While the authors' response did not strongly counter this criticism, this limitation could be considered acceptable given the relatively limited studies on random-walk-based graph encoders and the strong empirical performance demonstrated.

---

### Decision · Program_Chairs · 2025-01-22

Accept (Poster)